# ITERATIVE CONVERGENT COMPUTATION IS NOT A USEFUL INDUCTIVE BIAS FOR RESNETS

## ABSTRACT

Recent work has suggested that feedforward residual neural networks (ResNets) approximate iterative recurrent computations. Iterative computations are useful in many domains, so they might provide good solutions for neural networks to learn. Here we quantify the degree to which ResNets learn iterative solutions and introduce a regularization approach that encourages learning of iterative solutions. Iterative methods are characterized by two properties: iteration and convergence. To quantify these properties, we define three indices of iterative convergence. Consistent with previous work, we show that, even though ResNets can express iterative solutions, they do not learn them when trained conventionally on computer vision tasks. We then introduce regularizations to encourage iterative convergent computation and test whether this provides a useful inductive bias. To make the networks more iterative, we manipulate the degree of weight sharing across layers using soft gradient coupling. This new method provides a form of recurrence regularization and can interpolate smoothly between an ordinary ResNet and a "recurrent" ResNet (i.e., one that uses identical weights across layers and thus could be physically implemented with a recurrent network computing the successive stages iteratively across time). To make the networks more convergent we impose a Lipschitz constraint on the residual functions using spectral normalization. The three indices of iterative convergence reveal that the gradient coupling and the Lipschitz constraint succeed at making the networks iterative and convergent, respectively. However, neither recurrence regularization nor spectral normalization improve classification accuracy on standard visual recognition tasks (MNIST, CIFAR-10, CIFAR-100) or on challenging recognition tasks with partial occlusions (Digitclutter). Iterative convergent computation, in these tasks, does not provide a useful inductive bias for ResNets.

## 1 INTRODUCTION

An iterative method solves a difficult estimation or optimization problem by starting from an initial guess and repeatedly applying a transformation that is known to improve the estimate, leading to a sequence of estimates that converges to the solution. Iterative methods provide a powerful approach to finding exact or approximate solutions where direct methods fail (e.g., for difficult inverse problems or solutions to systems of equations that are nonlinear and/or large).

Recurrent neural networks (RNNs) iteratively apply the same transformation to their internal representation, suggesting that they may learn algorithms similar to the iterative methods used in mathematics and engineering. The idea of iterative refinement of a representation has also driven recent progress in the context of feedforward networks. New architectures, based on the idea of iterative refinement, have allowed for the training of very deep feedforward models with hundreds of layers. Prominent architectures for achieving high depth are residual (ResNets; He et al., 2016a) and highway networks (Srivastava et al., 2015), which use skip connections to drive the network to learn the residual: a pattern of adjustments to the input, thus encouraging the model to learn successive refinements of a representation of the input that is shared across layers.

These architectures combine two ideas. The first is to use skip connections to alleviate the problem of vanishing or exploding gradients (Hochreiter, 1991). The second is to make these skip connections fixed identity connections, such that the layers learn successive refinement of a shared repre-

sentational format. The second idea relates residual and highway networks to RNNs and iterative methods. Learning a single transformation that can be iteratively applied is attractive because it enables trading speed for accuracy by iterating longer (Spoerer et al., 2019). In addition, a preference for an iterative solution may provide a useful inductive bias for certain computational tasks.

However, it is unclear whether ResNets indeed learn solutions akin to iterative methods and, if they do, whether this is a useful inductive bias. The two defining features of iterative methods are (1) iteration and (2) convergence. Here we analyze to what extent these features emerge in ResNets. In order to investigate whether these features provide a useful inductive bias, we introduce two simple modifications of classical ResNets and study their impact on a number of datasets. First, we study CIFAR-10, CIFAR-100 (Krizhevsky, 2009), and MNIST (LeCun et al., 2010) as examples of classical vision tasks, assessing the networks' performance and sample efficiency. Since iterative and convergent inductive biases may be more useful for tasks that require some degree of recurrence, we also assess the networks' performance and sample efficiency on several variations of *Digitclutter*, a task which requires the recognition of multiple digits that occlude each other (Spoerer et al., 2017).

To study the effect of iteration, we manipulate the degree of weight sharing in ResNets, smoothly interpolating between ordinary and recurrent ResNets. We find that a higher degree of weight sharing tends to make the network more iterative, but does not result in improved performance or sample efficiency. This suggests that in ordinary ResNets, recurrent connections do not provide a useful inductive bias and the networks can harness the additional computational flexibility provided by non-recurrent residual blocks.

Recurrence implies iteration, but not convergence, and so is not sufficient for a network to implement an iterative method as defined above. ResNets, whether they are recurrent (i.e. sharing weights across layers) or not, are therefore neither required nor encouraged to converge during training. We demonstrate empirically that ResNets in general do not exhibit convergent behavior and that recurrent ResNets are more convergent than non-recurrent networks. To study the effect of convergence, we upper bound the residual blocks' Lipschitz constant. This modification adversely impacts performance, suggesting that the non-convergent behavior in ordinary ResNets is not merely due to lack of incentive, but underpins the networks' high performance. Across convergent ResNets, a higher degree of weight sharing does not negatively affect performance. This suggests that convergent ResNets, in contrasts to non-convergent ones, do not benefit from the increased computational flexibility of non-recurrent residual blocks.

Taken together, our results suggest that an inductive bias favoring an iterative convergent solution does not outweigh the computational flexibility of non-recurrent residual blocks for the considered tasks.

## 2 RELATED WORK

Prior theoretical work has focused on explaining the success of ResNets (He et al., 2016a) and the more general class of highway networks (Srivastava et al., 2015) by studying the learning dynamics in ResNets (Hochreiter, 1991; Orhan & Pitkow, 2018; Balduzzi et al., 2017) and their interpretation as an ensemble of shallow networks (Veit et al., 2016; Huang et al., 2018), as a discretized dynamical system (E, 2017; Haber & Ruthotto, 2018; E et al., 2019), and as performing iterative refinement.

**The iterative refinement hypothesis.** Our work builds on Jastrzebski et al. (2018) who argue that the sequential application of the residual blocks in a ResNet iteratively refines a representational estimate. Their work builds on observations that dropping out residual blocks, shuffling their order, or evaluating the last block several times retains reasonable performance (Veit et al., 2016; Greff et al., 2017) and can be used for training (Huang et al., 2016). Another set of methods uses such perturbations to train deep neural networks, using stochastic depth (Huang et al., 2016; Hu et al., 2019; Press et al., 2020). Other methods learn to evaluate a limited number of layers that depends on the input (Graves, 2017; Figurnov et al., 2017) or learn increasingly fine-grained object categories across layers (Zamir et al., 2017). Instead of using perturbations to encourage stability of the trained network, Ciccone et al. (2018) propose a method inspired by dynamical systems theory to guarantee such stability in their model.

**Iterative refinement and inverse problems.** The iterative refinement hypothesis is particularly important in the context of inverse problems, which are often solved using iterative methods. This is

particularly relevant for ResNets trained for perceptual tasks since perception is often conceptualized as an inverse problem (Poggio et al., 1985; Pizlo, 2001). Rao & Ballard (1999) modeled visual cortex using recurrent neural networks that iteratively infer the latent causes of a hierarchical Bayesian model. Though these networks have been applied to complex datasets (Wen et al., 2018), most models either learn the inverse model (Rezende et al., 2014; Kingma & Welling, 2013) or define an analytically invertible forward model (Rezende & Mohamed, 2015; Gomez et al., 2017). A notable exception are invertible ResNets (Behrmann et al., 2019), whose inverse can be computed through a fixed point iteration. Another set of works starts out with a classical iterative algorithm and unfolds this algorithm to create a deep network, approaching a similar problem from the opposite direction (Gregor & LeCun, 2010; Hershey et al., 2014; Wisdom et al., 2016; 2017). Moreover, Nguyen et al. (2019) refine CNNs to yield an inference algorithm on a specific generative model (recently implemented by Huang et al., 2020).

**Recurrent and residual neural networks.** The idea of iterative refinement has motivated an increasing number of recurrent convolutional neural networks being applied to computer vision without an explicit implementation of iterative inference (Kubilius et al., 2019; Spoerer et al., 2019). ResNets may be seen as RNNs that have been unfolded for a fixed number of iterations. Sharing weights between the different blocks allows us to train a recurrent residual neural network (Liao & Poggio, 2016). In this framework, recurrent residual neural networks may be seen as a special case of residual neural networks where the weights are equal between all blocks. Savarese & Maire (2019) and Oh et al. (2019) relaxed this constraint by defining the weights of the different blocks as a linear combination of a smaller set of underlying weights. The work by Savarese & Maire (2019) is particularly interesting in this context, as their method yielded a more efficient parameterization of Wide ResNets (Zagoruyko & Komodakis, 2017), which generally have more channels, but far fewer layers than the architectures we consider here.

**Inductive bias of recurrent operations on visual tasks.** The inverse problem perspective on perception suggests that ordinary recognition tasks in computer vision may already benefit from an iteratively convergent inductive bias. For other tasks, we have additional reasons to believe that recurrent processing may be beneficial. For example, object recognition in the primate ventral stream can be well captured by feedforward models (Hung et al., 2005; Majaj et al., 2015) but benefits form recurrent processing under challenging conditions. This includes tasks where the presented objects are partially occluded or degraded (Wyatte et al., 2014) or tasks that involve perceptual grouping according to local statistical regularities (Roelfsema et al., 1999) or object-level information (Vecera & Farah, 1997). These observations have inspired tasks such as Digitclutter and Digitdebris (Spoerer et al., 2017) as well as Pathfinder and cABC (Kim et al., 2020). For these datasets, adding recurrent networks have been shown to outperform corresponding feedforward control models (Spoerer et al., 2017; Linsley et al., 2020b)).

**Implicit recurrent computations.** Beyond potentially useful inductive bias, recurrent neural networks can provide additional computational advantages. If the recurrent operation converges to a fixed point, this fixed point can be determined more efficiently by classical iterative algorithms such as ODE solvers (Chen et al., 2018). Moreover, in this case, recurrent backpropagation (Pineda, 1987; Almeida, 1990) can compute the parameters' gradients much more efficiently than backpropagation through time (Werbos, 1988) because it does not require storing the intermediate activation. Its memory cost is therefore constant in depth. This method has inspired several promising models in computer vision. Deep equilibrium models (Bai et al., 2019) harness a quasi-Newton method to find the fixed point of their recurrent operation. They have recently been applied to object recognition and image segmentation tasks (Bai et al., 2020). These works empirically demonstrated the existence of a fixed point under their parameterization. Other models enforce such a fixed point by the Lipschitz constraints we here employ as well. Ciccone et al. (2018) use an upper bound based on the Frobenius norm, whereas Linsley et al. (2020a) approximate the Lipschitz constant using a vector-Jacobian product. Our method (see below) is based on Yoshida & Miyato (2017) and has recently been employed (for different purposes) by Miyato et al. (2018) and Behrmann et al. (2019).

## 3 DO RESNETS IMPLEMENT ITERATIVE COMPUTATIONS?

Iterative methods are characterized by two key properties: iteration (i.e., recurrent computation) and convergence. In this section, we examine whether residual neural networks have these proper-

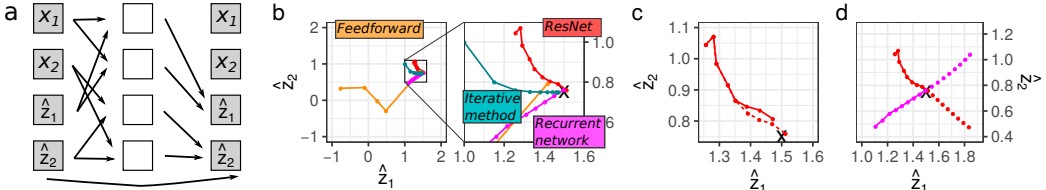

Figure 1: **a** A recurrent ResNet implementing a simple error-corrective inverse model. The prediction based on the current estimate $\hat{z}$ is compared to the input $x$ (via positive and negative errors $p$ and $n$). The error is used to update $\hat{z}$, whereas $x$ is simply retained throughout all blocks. **b** Trajectories of the estimates $\hat{z}_1$, $\hat{z}_2$ across blocks in a feedforward network, iterative steps in an the error-corrective algorithm, residual blocks in a trained ResNet, and residual blocks in a trained recurrent ResNet. All four methods converge to the correct estimate (indicated by black 'x'). **c** Dropping out the fourth block (unbroken line) has a minor impact on the ResNet. **d** If the last block is iteratively applied to the final estimate, the value diverges for both the residual and the recurrent network (broken lines), indicating that they do not learn a convergent structure.

ties. We start by showing that ResNets *can express iterative algorithms*, but then demonstrate that ResNets *do not automatically learn iterative algorithms*. We show that this observation extends to large-scale neural networks trained on real data. Finally, we introduce a paradigm that allows us to compare neural networks to iterative methods.

## 3.1 RESNETS CAN REPRESENT ITERATIVE COMPUTATIONS

A popular application for iterative methods is given by inverse problems of the form $x = f(z)$. It is often not possible to directly compute the inverse, $f^{-1}$, of the forward model, because of analytical or combinatorial intractability. Instead, approximations (Gershman & Goodman, 2014) or iterative error-corrective algorithms (Donoho, 2006) are used. Consider the linear forward model $x = f(z) := (\alpha_1 z_1, \alpha_2 z_2)$. Though $f$ has an analytical inverse, it serves as an illustrative example. Based on an input $x$, we can infer the latent variable $z$ using the iterative update

$$\hat{z}^{(0)} := x, \quad \hat{z}_i^{(t+1)} := \hat{z}_i^{(t)} - s_i \cdot \epsilon_i \cdot (\alpha_i \hat{z}_i^{(t)} - x_i), \quad s_i := \text{sign}(\alpha_i),$$

where $\hat{z}^{(T)}$, for some $T$, is the estimate of $z$ and $\epsilon_i > 0$ should be sufficiently small[1].

This update can be implemented in a small ResNet. Figure 1a contains a schematic illustration of one block of this network. The representation spanned by an encoder in the beginning of the network contains an input representation $x$ and a representation of the current estimate $z$. In the hidden layer of the residual block, we determine the positive and negative prediction errors and use them to update $z$:

$$p_i^{(t)} := \text{ReLU}(\alpha_i z_i^{(t)} - x_i), \quad n_i^{(t)} := \text{ReLU}(x_i - \alpha_i z_i^{(t)}), \quad z_i^{(t+1)} := z_i^{(t)} - s_i \epsilon_i p_i^{(t)} + s_i \epsilon_i n_i^{(t)}.$$

This recurrent residual building block would implement an iteratively convergent computation in a ResNet. The linear model is, of course, a trivial case, but serves as an illustration of the appeal of this approach. A wide neural network is a flexible function approximator and learning to represent the prediction error instead of the prediction is easier in many cases (Zamir et al., 2017).

## 3.2 RESNETS DO NOT AUTOMATICALLY LEARN ITERATIVE COMPUTATIONS

The fact that ResNets can express iterative computations does not imply that they necessarily learn iterative solutions when trained via backpropagation.

Here we consider the simple example from above to better understand the behavior that may emerge. We highlight three behaviors that distinguish iterative from non-iterative computations and will examine the behavior of large-scale neural networks in the following sections. As an example, we

---

[1]In the case of a linear function, any $\epsilon_i \leq 1$ works. The example given in figure 1 uses $\epsilon_1 = 0.3$, $\epsilon_2 = 0.8$.

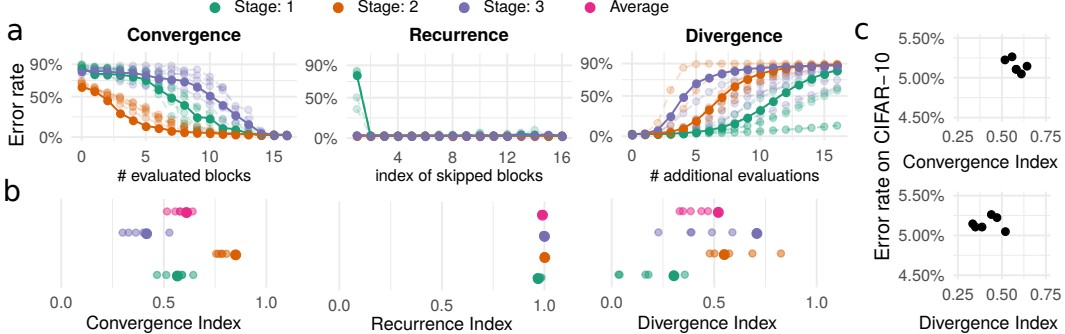

Figure 2: Iterative convergence in ResNets with standard training. **a** The different perturbation methods (early read-out for determining convergence, dropping-out blocks for determining recurrence, and additional evaluations of the last block for determining divergence) are illustrated for the three stages of the ResNet. The x axis depicts the residual block targeted by the perturbation and the y axis the error rate resulting from the corresponding perturbation (chance performance at 90%). For clarity, one of the six instances is emphasized in the plots. **b** The resulting index values for each stage (small translucent dots) and their averages across instances (large dots). **c** The error rate for the individual network instances is plotted against Convergence and Divergence Index.

train a conventional ResNet, and a ResNet that uses the same weights in each block (equivalent to a recurrent network) to invert the function $f(z) := (\frac{3}{2}z_1, \frac{3}{4}z_2)$. We contrast their behavior with the iterative error-corrective algorithm outlined above.

Due to the lack of constraints, a non-residual feedforward neural network changes its representation in every layer. As a consequence, the linear decoder at the end of the network is not aligned to the intermediate representation and early readout (as depicted by the orange dots in Fig. 1b) leads to a meaningless estimate. In contrast, the skip connections encourage a ResNet to use the same representational format across blocks. This is to say that its intermediate representations are better aligned with the final decoder. Early readout is therefore possible and the representation across blocks will approach the final estimate. As a consequence, the across-block dynamics of the non-residual network are meaningless, whereas the recurrent and residual network's early readouts are close to the final estimate and approach it in a smooth manner, just like the error-corrective algorithm (Figure 1b). Since iterative methods iteratively refine their initial estimate, their behavior is more similar to the ResNets' monotonic convergence.

Aside from their smooth convergence, a fundamental property of iterative methods is their recurrence (i.e., the repeated use of the same computation). This means that dropping out an earlier block has the same effect as dropping out the last one. We can relax the requirement for exact repetition (weight sharing) and require merely similar computations. Figure 1c illustrates that the trained ResNet is indeed relatively robust to block dropout.

Yet the learned models fall short of an iterative method, which is apparent from a third mode of investigation. Iterative convergence would imply that applying the last block's transformation iteratively should keep the readout in the vicinity of the actual estimate. This is clearly not the case in our toy example (Figure 1d). Rather than representing a convergent estimate, this result is more compatible with understanding ResNets as approaching their final estimate at a constant speed and, in the case of late readout, moving past this estimate and overshooting at the same speed. Notably, both the non-recurrent and recurrent networks exhibit this behavior.

### 3.3 ITERATIVE CONVERGENCE INDICES

This behavior is not surprising. After all, the network is trained to work for a fixed number of steps and not constrained to stay within the vicinity of its final estimate if more steps are added. However, it reveals that even recurrent ResNets do not automatically learn iterative convergent computations. To assess the extent to which they do learn iterative convergent computations we define three continuous indices, which measure convergence, recurrence, and divergence (defined below).

We evaluate the indices for six instances of ResNet-101, trained on CIFAR-10 (Krizhevsky, 2009). This ResNet consists of three stages of 16 residual blocks with 16, 32, and 64 channels, respectively.[2] The ResNet achieved 5.2% classification error on the validation set. To characterize the extent to which the ResNets have learned an iterative convergent computation, we introduce three indices measuring different aspects of such computations. [3]

**Convergence Index.** Viewing ResNets as performing iterative refinement suggests that each stage gradually approaches its final estimate before passing this estimate to the next stage using a down-sampling layer. By passing the estimate at each of the residual blocks to the next stage, we can monitor how the stages approach their final estimate across blocks (see Figure 2a, left panel). In accordance with previous results (Jastrzebski et al., 2018), we find that all stages smoothly approach their final estimate, confirming the earlier intuition of a shared representational format. To measure the rate of convergence, we compute the area under the curve (AUC) of the classification error, which we call the Convergence Index. We invert and normalize this value such that a Convergence Index of 0 corresponds chance level read-out at each residual block, whereas a Convergence Index of 1 corresponds to an instant convergence to the final classification error at the first residual block. Figure 2b, left panel, depicts this value for each stage, and averaged across stages.

**Recurrence Index.** To measure the degree of recurrence, we evaluated the effect of dropping out individual blocks on the error rate of the network (see Figure 2a, middle panel). In a non-recurrent ResNet, dropping out earlier blocks may have a stronger effect on the error rate than dropping out the last block. In contrast, in a recurrent ResNet, the effect on error rate is the same for dropping out either earlier blocks or the last block. We therefore computed the difference in error rate observed after these two manipulations. We summarized the behavior by the AUC, which we refer to as Recurrence Index (RI). We invert and normalize this value such that the RI is 0 if dropping out any block leads to an error rate at chance level and the RI is 1 in the case of a recurrent algorithm. Even though we study non-recurrent ResNets, dropping out any block other than the very first leads to a negligible drop in performance, replicating previous results (Jastrzebski et al., 2018; Veit et al., 2016). As a consequence, the RI, across all stages, is close to 1.0 (see Figure 2b, middle panel).

**Divergence Index.** ResNets may either converge to their final estimate or simply approach it in a sequence of steps. An iterative algorithm should not be negatively affected by additional applications of the same function. To examine this property, we apply the last block of each stage for an additional up to sixteen steps (see Figure 2a, right panel) and determine the AUC (Divergence Index, DI, see Figure 2b, right panel). We find that no stage is particularly robust to such additional evaluations, though the first stage has the lowest DI, indicating that it is the most robust. This suggests that ResNets approach and move away from their final estimate in a sequence of steps, with their computations bearing little similarity to an iterative convergent algorithm. A high DI does not indicate that the ResNet has failed in some way. After all, it was not trained to be robust to such perturbations. However, it indicates that the ResNet may not implement an iterative convergent computation.

## 4    MANIPULATING CONVERGENCE AND ITERATION IN RESIDUAL NETWORKS

We provided indices measuring convergence, recurrence, and divergence to assess the degree to which a ResNet implements an iterative method. Even though they are able to, ResNets do not necessarily learn to implement a purely iterative method. In particular, they show divergent behavior. Nevertheless, as we have shown above, their behavior does show some similarity to iterative methods and their success has been attributed to these similarities (Greff et al., 2017; Jastrzebski et al., 2018). This suggests that even though the parameterization and optimization does not promote the emergence of an iterative method in a ResNet, a ResNet with iteratively convergent behavior may still have a better inductive bias. To test this hypothesis, we therefore here control the inductive bias, namely recurrence and convergence, of ResNets.

---

[2]The ResNet-101 uses the architecture recommended by He et al. (2016b). Appendix A includes details on the architecture and the training paradigm and Appendix F corresponding results on two further datasets, MNIST (LeCun et al., 2010) and Digitclutter-3 (Spoerer et al., 2017).

[3]Section B contains several variations on these definitions

## 4.1 SOFT GRADIENT COUPLING CAN INTERPOLATE SMOOTHLY BETWEEN ORDINARY AND RECURRENT OPTIMIZATION

We propose a method to blend between recurrent and non-recurrent networks without changing the architecture or the loss landscape. The method is motivated by the observation that we can train a recurrent neural network by sharing the different blocks' gradients (Jastrzebski et al., 2018; Liao & Poggio, 2016). In ordinary ResNets, the residual block $t$ with the weights $\boldsymbol{W}_t$ is changed by following the gradient $\Delta_t = \partial_{\boldsymbol{W}_t} L$, whereas RNNs impose as gradient

$$\Delta = T^{-1} \sum_{t=1}^{T} \partial_{\boldsymbol{W}_t} L, \tag{1}$$

where the weights across residual blocks within a stage must start from the same initialization. The former means that we do not employ any inductive bias towards recurrence, whereas the latter imposes a possibly overly restrictive function space on the architecture. To address both limitations, we propose **soft gradient coupling**, which uses as its update rule

$$\tilde{\Delta}_t = (1 - \lambda)\Delta_t + \lambda\Delta, \quad \lambda \in [0, 1]. \tag{2}$$

For $\lambda < 1$, this retains the entire space of computations enabling both non-recurrent as well as recurrent computations. However, for $\lambda > 0$, the optimization is biased to find more recurrent optima. In contrast to penalty regularizations or strict weight sharing models (Oh et al., 2019; Savarese & Maire, 2019), this does not change the network or loss landscape, but simply the accessibility of different local minima of the loss landscape.

For networks with coupled gradients (i. e., $0 < \lambda \leq 1$), we initialize their weights recurrently (i.e., all residual blocks within one stage share the same initialization). We unshare batch norm statistics as suggested by Jastrzebski et al. (2018), but leave their parameters (softly) coupled.

## 4.2 SPECTRAL NORMALIZATION CAN GUARANTEE CONVERGENCE IN RESIDUAL NETWORKS

Iterative methods preserve a stable output when applied repeatedly. In contrast, the output of the ResNets diverged when the last block was applied repeatedly beyond the number of steps it was trained for. In order to control the degree of convergence in a ResNet, we constrain the Lipschitz constant $L$ of the residual function $f$. $L$ is defined as the minimal value such that for any input $\boldsymbol{x}, \boldsymbol{y}$, $\|f(x) - f(y)\| \leq L\|x - y\|$. The smaller $L$, the more stable $f$. The Lipschitz constant is hard to determine accurately as it is a global property of $f$. We therefore determine an upper bound $\hat{L}(f) \geq L$ based on the linear operations within $f$ (see section E.1). Using this upper bound, we replace $f$ by its **spectral normalization**

$$\tilde{f}(\boldsymbol{x}) := f(\boldsymbol{x})/c, \quad c := \max(\mu/\hat{L}(f), 1), \tag{3}$$

for a certain value $\mu$. If $\hat{L}(f) \leq \mu$, the corrective factor $c$ will not change the function. If $\hat{L}(f) > \mu$, it will set the corresponding upper bound of $\tilde{f}$ at $\hat{L}(\tilde{f}) = \mu$, constraining the residual function's Lipschitz constant.

For a given input $\boldsymbol{x}$, a recurrent residual stage is defined by the iterative application $\boldsymbol{z}_0 := \boldsymbol{x}$, $\boldsymbol{z}_t := R(\boldsymbol{z}_{t-1})$, where $R$ defines the recurrently applied residual block. We wish to guarantee that $\boldsymbol{z}_t$ eventually converges to a fixed point $\boldsymbol{z}_\infty$ and hope that empirically, $\boldsymbol{z}_T$, the representation after the specified number of iterations, will be close to this fixed point. According to the Banach fixed point theorem, one way to guarantee such convergence is to require that the residual block's Lipschitz constant $L_R$ be smaller than one.

We can achieve this by replacing the residual connections between adjacent blocks by residual connections between the input $\boldsymbol{x}$ to the stage and each block, i. e. $\tilde{R}(\boldsymbol{z}) := \boldsymbol{x} + \tilde{f}(\boldsymbol{z})$. $\tilde{R}$ has the same Lipschitz constant as $\tilde{f}$. Setting $\mu < 1$, we therefore guarantee that $\tilde{R}$ converges to a fixed point defined by $\boldsymbol{x}$. We call this network the *properly convergent ResNet* (PCR).

To get a network more similar to an ordinary ResNet, we consider $R(\boldsymbol{z}) := \boldsymbol{z} + \tilde{f}(\boldsymbol{z})$. Though we convergence is not guaranteed for the network defined by this residual block, we show its empirical convergence. We thus call this network the *improperly convergent ResNet* (ICR).

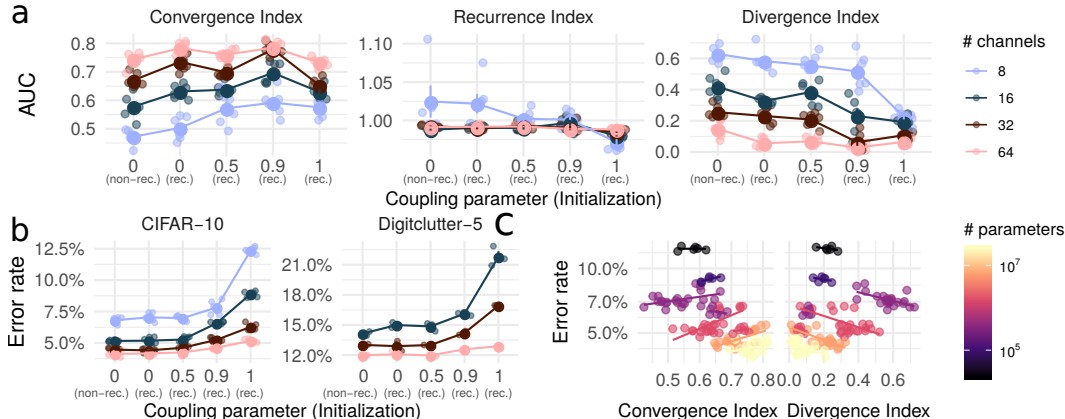

Figure 3: Iterative convergence and performance of coupled ResNets. **a** Effect of gradient coupling and initialization (rec.: recurrent; non.-rec.: non-recurrent) on indices of iterative convergence for architectures with different numbers of channels. **b** Effect of gradient coupling and initialization on the performance on CIFAR-10 and Digitclutter-5. **c** Relationship between performance and iterative convergence, i.e., Convergence (left) and Divergence Index (right). Models with the same number of parameters are visualized by the same color and individual lines. Results in **a**, **c** are on CIFAR-10.

## 5 EXPERIMENTS

To assess our hypotheses, we considered non-recurrently initialized (ordinary) ResNets as well as recurrently initialized ResNets with coupling parameters 0, 0.5, 0.9, and 1. In addition, we considered properly and improperly convergent ResNets across the same coupling parameters, setting $\mu = 0.95$ (we only trained these networks on CIFAR-10). We trained several instances of all these ResNets with 8, 16, 32, and 64 channels in the first stage on classical visual recognition tasks (CIFAR-10, MNIST, and CIFAR-100) as well as *Digitclutter*, a challenging task with partially occluded objects, which has previously been observed to benefit from recurrent connections (Spoerer et al., 2017).

### 5.1 SOFT GRADIENT COUPLING IMPROVES ITERATIVE RECURRENCE INDICES

As Figure 3a shows, soft gradient coupling indeed improves iterative convergence, increasing the Convergence Index and decreasing the Divergence Index. The Recurrence Index is centered closely around 1. Convergence and Divergence Index, on the other hand, tend to increase and decrease, respectively, both with higher coupling parameters and with a higher number of channels. Notably, this trend appears to not hold up for a fully recurrent ResNet, corresponding to a coupling parameter of 1. These results show that soft gradient coupling is an effective way of manipulating a ResNet's behavior. Increasing weight similarity across blocks leads to more iterative convergence in a ResNet.

### 5.2 SPECTRAL NORMALIZATION MAKES RESNETS CONVERGENT

Both the properly and improperly convergent ResNets have a high Convergence Index as well as a Divergence Index at almost zero (see Fig. 4a for ICRs and section E.2 for PCRs). The Recurrence Index is again centered around one. For improperly convergent ResNets with 16 or 32 channels in the first stage, higher coupling parameters generally have a lower Convergence Index and a higher Divergence Index. Nevertheless, these indices indicate that the spectrally normalized ResNets exhibit much more convergent behavior than the ordinary networks. This was only guaranteed for the recurrent, properly convergent ResNets and is therefore an important observation.

### 5.3 STRONGER ITERATIVE CONVERGENCE DOES NOT PROVIDE A USEFUL INDUCTIVE BIAS

We first assessed the effect of gradient coupling on the performance of non-convergent ResNets. As Fig. 3b shows, a higher coupling parameter consistently leads to a higher error rate, both for CIFAR-10 and Digitclutter. Additional supporting experiments on CIFAR-100, MNIST, the Digit-

Figure 4: Iterative convergence and performance of improperly convergent ResNets. **a** Effects of gradient coupling and initialization (rec.: recurrent; non.-rec.: non-recurrent) on iterative convergence indices. **b** Error rates on CIFAR-10 as a function of gradient coupling and initialization.

clutter task, and on sample efficiency can be found in section F. However, for intermediate coupling parameters of 0.5 and 0.9, this increase in error rate is smaller for networks with higher capacity (i.e., more channels). This effect can also be seen from the relationship between iterative convergence indices and performance. In particular, Figure 3c demonstrates that performance is higher for ResNets with a higher Convergence Index and a lower Divergence Index. This effect, however, is driven by the fact that ResNets with a higher number of channels also show higher measures of iterative convergence (see Figure 3a). When controlling for the number of parameters (see lines in Figure 3c), we find no clear relationship between Convergence and Divergence Index and performance.

We then assessed the effect of convergence regularization on performance by training several convergent ResNets on CIFAR-10 (see Fig. 4b for ICRs). Convergence regularization led to a higher error rate across all coupling parameters and architectures. A notable exception is the fully coupled ResNet with 16 channels, which performs equally with and without convergence regularization. This suggests that convergence is not a useful inductive bias in ResNets. Taken together, these experiments suggest that iterative convergence may not provide a useful inductive bias for ResNets.

## 6 DISCUSSION

We introduced soft gradient coupling, a new method of recurrence regularization, and demonstrated that this enabled us to manipulate iterative convergence properties in ResNets. To measure these properties, we introduced three indices of iterative convergence, quantifying the effect of perturbations previously introduced in the literature (Veit et al., 2016; Jastrzebski et al., 2018).

Iterative methods are considered powerful approaches in particular for solving difficult inverse problems. However, here we did not find iterative convergence to be a useful inductive bias for ResNets. Moreover, we found that higher degrees of weight sharing did not improve a ResNet's parameter efficiency. One reason for this may be that soft gradient coupling or the spectral normalization are the wrong methods for this purpose or require a different optimization strategy. Our findings also suggest, however, that deep feedforward computations ahould perhaps not be characterized as iterative refinement on a latent representation, but simply as a sequence of operations smoothly approaching their final estimate. Our conclusions are based on experiments on four visual classification tasks. Visual tasks have been proposed to be inverse problems and therefore lend themselves to iterative inference algorithms. Recognition tasks like Digitclutter that involve partial occlusions of objects have in particular been shown to benefit from recurrent computations (Wyatte et al., 2014; Spoerer et al., 2017). However, an iterative method like an error-corrective algorithm that would require a forward model of the data may be more complex and therefore harder to learn than a purely discriminative model. Hence, for the four tested tasks, ResNets may learn direct inference rather than error-corrective inference via a forward model.

Although it did not improve performance here, soft gradient coupling provides a method for smoothly interpolating between feedforward and recurrent neural networks. More generally, soft gradient coupling provides a simple way to encourage sets of weights to remain similar to each other. This technique may find further use in relaxing weight-sharing constraints and studying the benefit of various forms of weight sharing, including recurrence and convolution, in deep neural networks.

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

# A NETWORK TRAINING

## A.1 SOFTWARE

We trained the networks using PyTorch (Paszke et al., 2019), PytorchLightning, numpy (Oliphant, 2006), and pandas in Python (Van Rossum & Drake, 2009). The remaining analysis was conducted in R (R Core Team, 2019), using ggplot2 (Wickham, 2016), dplyr (Wickham et al., 2019), tidyr (Wickham & Henry, 2019), patchwork (Pedersen, 2019), and DescTools (Signorell, 2020). The implementation of ResNet-104 was significantly supported by an existing implementation (Idelbayev, 2018).

## A.2 DATASETS

We trained and evaluated the networks on four datasets: CIFAR-10, CIFAR-100 (Krizhevsky, 2009), MNIST (LeCun et al., 2010), and Digitclutter (Spoerer et al., 2017). All images were normalized before being provided to the network. For CIFAR-10, we used a training/validation sample split of 45000/5000, augmented the data by random cropping and random horizontal flips during training. For MNIST, we used a split of 50000/10000, and augmented the data by random cropping. Finally, for Digitclutter, we used between two and five overlapping digits (referred to as Digitclutter-3) with a split of 100000/10000 without data augmentation.

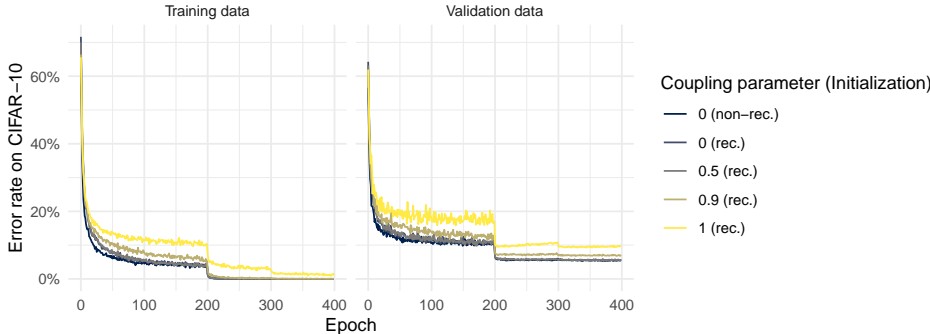

Figure 5: This figure depicts the evolution of mean error rate on training (left) and validation data (right) during training. This pertains to 16 channels in the first stage across all combinations of coupling parameter and initialization.

## A.3 ARCHITECTURE DETAILS

As our base architecture we use a ResNet with the preactivation unit recommended by He et al. (2016b). This means that a residual block consists of

$$BN_1 \rightarrow ReLU \rightarrow Conv_1 \rightarrow BN_2 \rightarrow ReLU \rightarrow Conv_2,$$

where BN stands for batch normalization (Ioffe & Szegedy, 2015), ReLU for rectified linear unit, and Conv for a convolutional layer.

For CIFAR-10, CIFAR-100, and Digitclutter, this ResNet contained three stages with 16 blocks each. This is a common depth for a ResNet. For example He et al. (2016a) use a ResNet-110. Two stages were connected by a downsampling block, which halved width and height of the representation and doubled the number of channels. The downsampling block consisted of a residual block where the first convolutional layer had stride 2 and doubled the number of channels. The shortcut connection (otherwise simply the identity function) decreased the image resolution by taking into account only every second column and row. To increase the number of channels, the shortcut connection added a number of layers initialized as zero. This downsampling block was never removed as part of the perturbations defining the indices of iterative convergence.

This means that for a standard ResNet with 16 channels in the first stage, the second stage consisted of (16x16)-representations with 32 channels, and the third stage consisted of (8x8)-representations with 64 channels. In contrast, we only used one stage with 16 blocks for MNIST.

## A.4 TRAINING DETAILS

We initialized our networks using Kaiming initialization (He et al., 2015) for the convolutional and linear weights. In the case of a recurrent initialization, these were equal across residual blocks of a stage. In the case of a non-recurrent initialization, they were drawn independently for each block. We initialized the batch normalization's scale $\gamma$ with 1 and its bias with 0.

We trained the original networks for 400 epochs using gradient descent with momentum 0.9. For CIFAR-10, we used an initial learning rate of 0.1, which was divided by 10 after 200 and 300 epochs (see figure 5 for the evolution of the learning rate of the ResNets with 16 channels in the first stage). For CIFAR-100 and Digitclutter, we trained the networks for 200 epochs and used an initial learning rate of 0.1, which was divided by 10 after 100 and 150 epochs. For MNIST, we used an initial learning rate of 0.025, which was divided by 2.5 after 200 and 300 epochs. For Digitclutter, we used an initial learning rate of 0.05, which was divided by 5 after 200 and 300 epochs. We then identified the epoch at which the network had obtained the best validation classification error and used the corresponding model for all analyses.

## B   ALTERNATIVE INDICES OF ITERATIVE CONVERGENCE

We have defined our indices of iterative convergence as based on the accuracy with respect to the true image labels. This makes sense because computer vision models are usually evaluated on accuracy and we may not care about perturbations which do not affect accuracy. Nevertheless, only studying the perturbations' effect on accuracy may lead us to miss certain phenomena. For example, an intermediate representation may have a similar accuracy as the unperturbed model, but make largely inconsistent predictions. This would be important to know. To address this limitation, we studied the accuracy of intermediate representations with respect to the predictions made by the unperturbed model instead of the ground truth. Fig. 6a depicts the indices of iterative convergence for the non-convergent ResNets trained on CIFAR-10. As we can see, this alternative definition does not change the conclusions we can draw from the experiment.

## C   PARAMETER EFFICIENCY

### C.1   EFFECTIVE PARAMETER COUNT

Softly coupled ResNets have the same number of raw parameters as ordinary ResNets with the same architecture. Their soft weight sharing, however, increases the similarity between weights of different blocks. This weight similarity is not reflected in the raw parameter count. We therefore introduce an **effective parameter count** (EPC). For this measure, three quantities are of interest: the parameters that are not coupled with any other parameters, $\boldsymbol{W}_{\text{uncoupled}}$, the mean parameters for a given stage $s$, $\overline{\boldsymbol{W}}^{(s)}$, and the normalized deviations from this mean of a parameters within a particular layer, $\boldsymbol{D}_l^{(s)} := (\boldsymbol{W}_l^{(s)} - \overline{\boldsymbol{W}}^{(s)})/\hat{\sigma}(\overline{\boldsymbol{W}}^{(s)})$. As a measure of the effective number of parameters which is sensitive to different degrees of weight sharing, we now propose

$$\text{EPC}(\boldsymbol{W}) := \|\boldsymbol{W}_{\text{uncoupled}}\|_0 + \sum_{s=1}^{S} \|\overline{\boldsymbol{W}}^{(s)}\|_0 + \sum_{s=1}^{S}\sum_{l=1}^{L} \|\boldsymbol{D}_l^{(s)}\|_1, \qquad (4)$$

where the $L_0$-norm $\|\cdot\|_0$ corresponds to the parameter count, and $S$ and $L$ to the number of stages and blocks, respectively. This allows us to interpolate between the raw parameter counts of non-recurrent and recurrent ResNet. For softly coupled ResNets, the EPC measure lies between the raw parameter count of non-recurrent and recurrent ResNets as a function of the weight similarity across weights (see Fig. 7b).

This definition also allows us to examine the effect of gradient coupling on the average deviation of the parameters from the mean. As expected higher coupling parameter lead to a lower average deviation and $\lambda = 1$ leads to no deviation at all (see Fig. 7c).

### C.2   STRONGER GRADIENT COUPLING DOES NOT INCREASE PARAMETER EFFICIENCY

We hypothesized that ResNets may employ implicitly recurrent computations. This would mean that there is a more recurrent network which is more parameter-efficient. If this is the case, soft gradient coupling might be able to find such a solution. For a given number of parameters, however, recurrent and softly gradient-coupled networks do not outperform ordinary ResNets (see Figure 3c, left). Softly coupled networks even tend to do worse. This is because pure parameter counts do not take into account the soft weight sharing across blocks. We therefore developed a measure of the effective parameter count which takes the similarity of weights across layers into account (see Section C.1). Figure 7c, right, shows that for a given effective parameter count, ordinary, softly gradient-coupled, and recurrent ResNets all perform equally well. Nonetheless, this suggests that we cannot increase parameter efficiency by making the ResNets more recurrent.

## D   TRAINING VARIATIONS

On CIFAR-10, we explored several variations of the training algorithm, which were consistent with the findings presented above.

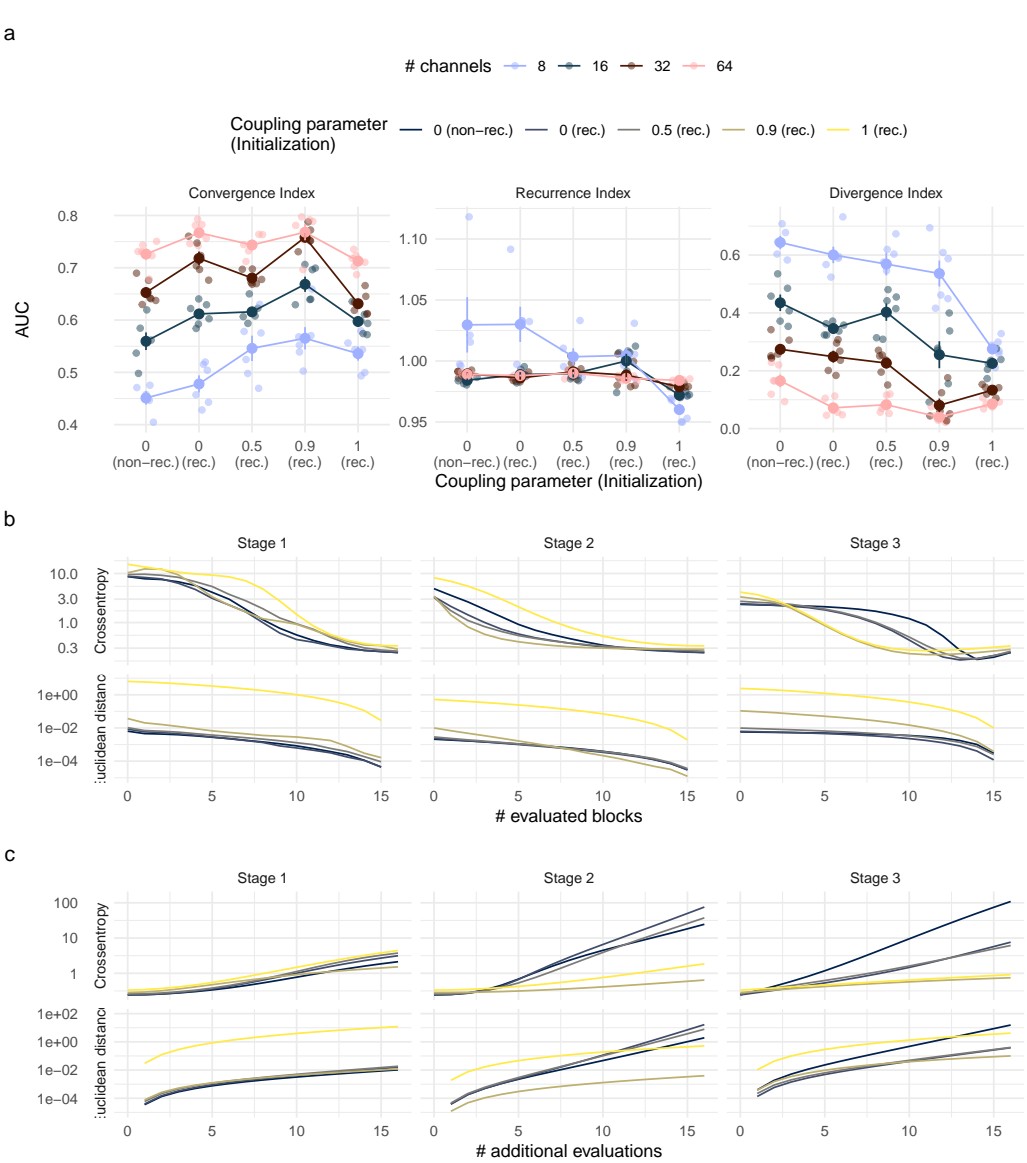

Figure 6: **a** The indices of iterative convergence as based on accuracy with respect to the predictions made by the unperturbed model rather than the ground truth. **b,c** The evolution of crossentropy and the Euclidean distance between intermediate representations and the final representation across the number of evaluated blocks and additional evaluations.

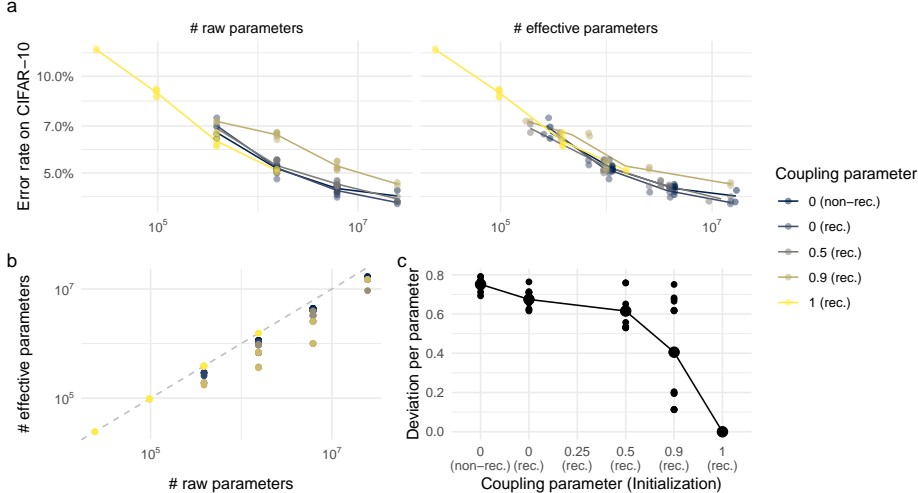

Figure 7: This figure plots the effective parameter count of the networks studied in the main text against the raw parameter count.

### D.1 RECURRENT BATCH NORMALIZATION

Cooijmans et al. (2017) proposed to initialize the scale of the batch normalization $\gamma$ as 0.1 in recurrent neural networks. When applying this method to softly gradient-coupled networks, we find that the method strongly improves performance for $\lambda = 0.9$, but not for any other coupling parameters (see figure 8a).

### D.2 TRIANGULAR GRADIENT COUPLING

We may generalize soft gradient coupling to the coupling rule

$$\tilde{\Delta}_t = \sum_{s=1}^{T} \kappa(s, t; \lambda) \partial_{\boldsymbol{W}_t} L, \tag{5}$$

where $\kappa$ is some kernel depending on the coupling parameter $\lambda$. Ordinary soft gradient coupling can be recovered with a uniform kernel

$$\kappa(s, t; \lambda) := \begin{cases} (1 - \lambda) + \lambda/T & \text{if } s = t, \\ \lambda/T & \text{if } s \neq t. \end{cases} \tag{6}$$

Alternatively, we may want to couple adjacent blocks more strongly than blocks that are far away from each other. This can, for example, be achieved with a triangular kernel

$$\kappa(s, t; \lambda) := \begin{cases} (1 - 2 \cdot (1 - \lambda)/T \cdot |s - t|)^+ & \text{if } \lambda \geq 0.5, \\ (1 - 1/(2 \cdot \lambda \cdot T) \cdot |s - t|)^+ & \text{if } \lambda < 0.5, \end{cases} \tag{7}$$

where

$$(\cdot)^+ = \max(\cdot, 0).$$

Again, $\lambda = 0$ corresponds to an ordinary ResNet and $\lambda = 1$ corresponds to a fully recurrent ResNet, whereas intermediate values for $\lambda$ smoothly interpolate between the two.

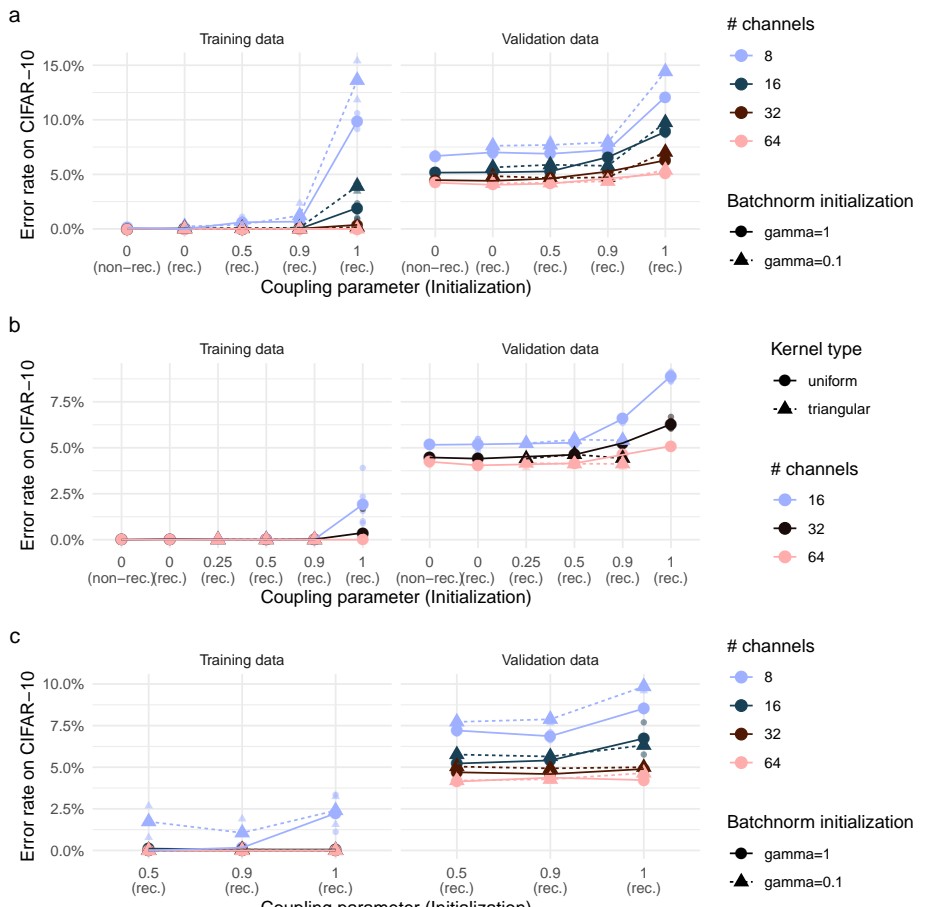

Figure 8: Performance of training variations on CIFAR-10. **a** The effect of initializing batchnorm with $\gamma = 0.1$ instead of $\gamma = 1$. **b** The effect of using a triangular kernel for gradient coupling instead of a uniform kernel. **c** A variation of gradient coupling where the first five blocks in each stage were uncoupled.

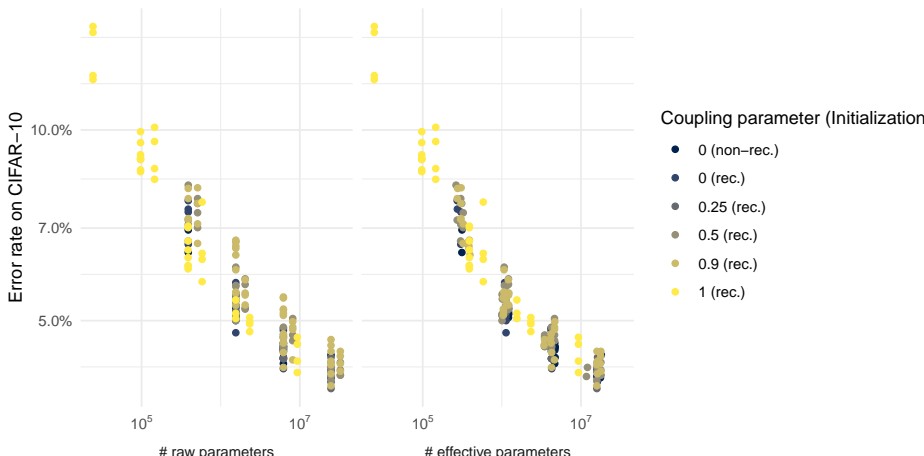

Figure 9: This figure plots the error rate of all models trained on CIFAR-10 against their raw and effective parameter count.

Figure 8b shows the effect of triangular as compared to uniform gradient coupling. For a given coupling parameter, this method appears to outperform the uniformly gradient-coupled network, but neither method outperforms an ordinary ResNet.

### D.3 START COUPLING AT LATER LAYERS

Finally, we uncoupled the first five residual blocks and only constrained the remaining blocks. For the fully recurrent case, this corresponds to the implementation by Jastrzebski et al. (2018). We also studied the effect of this late coupling together with a batchnorm initialization of $\gamma = 0.1$. In both cases, the late coupling means that the coupled networks are better than coupled networks which start coupling at block 0, but still worse than an ordinary ResNet.

### D.4 PARAMETER EFFICIENCY

Figure 9 shows that across all these extended networks, the relationship between effective parameter count and performance is also largely unaffected by coupling parameter.

### D.5 ITERATIVE CONVERGENCE INDICES

Figure 10 demonstrates that for the extended training variations, higher coupling parameters generally tend to increase the Convergence Index and decrease the Divergence Index, as well. The extended training variations also predominantly yield networks with a Recurrence Index around 1.

## E DETAILS ON THE SPECTRAL NORMALIZATION

### E.1 UPPER BOUND ON THE LIPSCHITZ CONSTANT

To determine an upper bound on the Lipschitz constant of $f$, we use an alternative characterization based on the Jacobian $J_f(x)$. The *spectral norm* $\|A\|_2$ of a matrix $A$ is defined as its maximal singular value. The Lipschitz constant is then given by

$$L(f) = \max_x \|J_f(x)\|_2.$$

According to the chain rule,

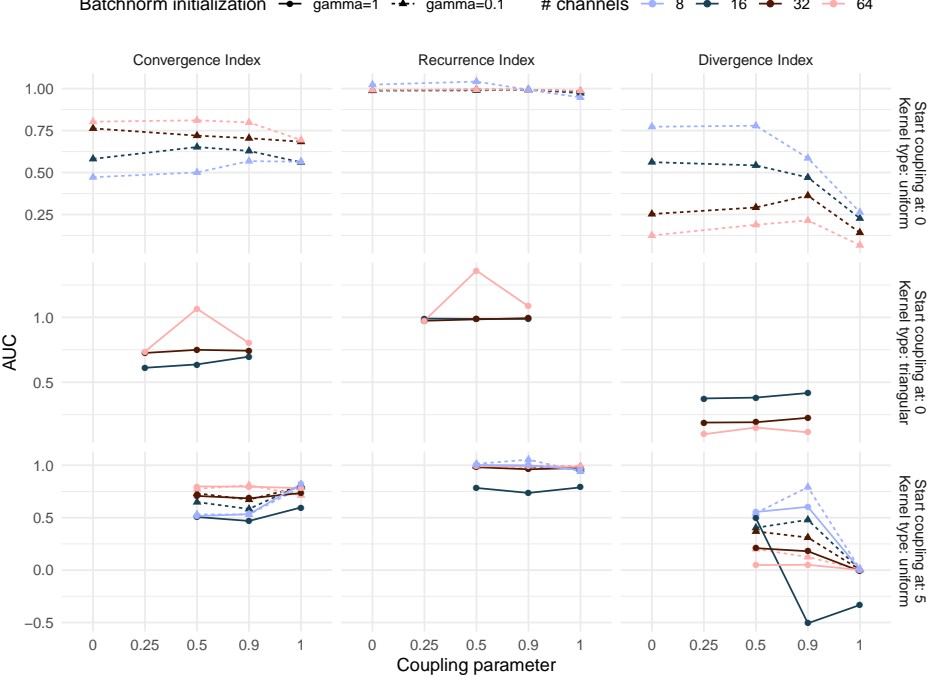

Figure 10: This figure plots the indices of iterative convergence against the coupling parameters for the different training variations.

$$J_f(x) = J_{\mathrm{Conv}_2} \cdot J_{\mathrm{ReLU}}(z_2) \cdot J_{\mathrm{BN}_2 \circ \mathrm{Conv}_1} \cdot J_{\mathrm{ReLU}}(z_1) \cdot J_{\mathrm{BN}_1}.$$

Here $z_1$ and $z_2$ are given by the representation at the appropriate intermediate stages of the residual function, i. e. $z_1$ is the representation after $\mathrm{BN}_1$ and $z_2$ is the representation after $\mathrm{BN}_2$. The Jacobians of the batch normalizations and convolutions do not depend on the input as these are linear operations. Since $J_{\mathrm{ReLU}}$ and $J_{\mathrm{BN}_1}$ are both diagonal matrices, they commute and therefore,

$$\begin{aligned} J_f(x) &= J_{\mathrm{Conv}_2} \cdot J_{\mathrm{ReLU}}(z_2) \cdot J_{\mathrm{BN}_2 \circ \mathrm{Conv}_1} \cdot J_{\mathrm{BN}_1} \cdot J_{\mathrm{ReLU}}(z_1) \\ &= J_{\mathrm{Conv}_2} \cdot J_{\mathrm{ReLU}}(z_2) \cdot J_{\mathrm{BN}_2 \circ \mathrm{Conv}_1 \circ \mathrm{BN}_1} \cdot J_{\mathrm{ReLU}}(z_1). \end{aligned}$$

We have therefore split the $J_f$ into a product of two constant functions and two Jacobian of the rectified linear unit.

The spectral norm $\| \cdot \|_2$ is known to be sub-multiplicative. This means that for matrices $A$ and $B$, $\|BA\|_2 \leq \|B\|_2 \|A\|_2$. Moreover, $J_{\mathrm{ReLU}}$, depending on whether its input is positive or negative is given by a diagonal matrix with ones or zeros on the diagonal. Its singular values are therefore at most 1. Putting this together, we can upper bound the Lipschitz constant as

$$\begin{aligned} L(f) = \max_x \|J_f(x)\|_2 &\leq \max_x \|J_{\mathrm{Conv}_2}\|_2 \|J_{\mathrm{ReLU}}(z_2)\|_2 \|J_{\mathrm{BN}_2 \circ \mathrm{Conv}_1 \circ \mathrm{BN}_1}\|_2 \\ &\leq \|J_{\mathrm{Conv}_2}\|_2 \|J_{\mathrm{BN}_2 \circ \mathrm{Conv}_1 \circ \mathrm{BN}_1}\|_2 =: \hat{L}(f). \end{aligned}$$

Theoretically, we could determine the maximal singular value of the convolution using a singular value decomposition. However, the singular value decomposition of such a large matrix is computationally expensive. Instead, Yoshida & Miyato (2017) lay out how the maximal singular value can be approximated using a power iteration (von Mises & Pollaczek-Geiringer, 1929). The only difference to their method consists in the fact that our first convolution additionally involves multiplying the input by the diagonal matrices given by the two batch normalizations.

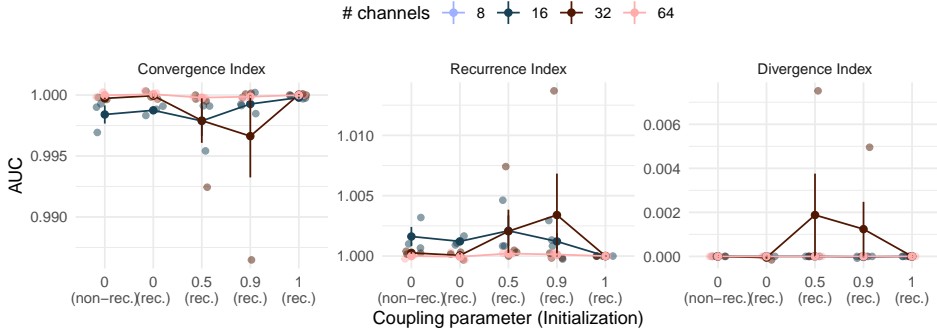

Figure 11: This figures plots the indices of iterative convergence for the properly convergent ResNets trained on CIFAR-10.

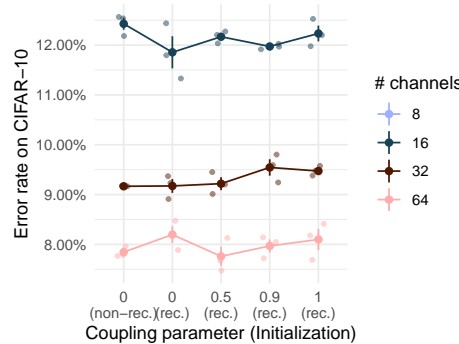

Figure 12: This figures depicts the performance of properly convergent ResNets on CIFAR-10.

## E.2 RESULTS ON PROPERLY CONVERGENT RESNETS

Fig. 11 demonstrates that the properly convergent ResNets indeed converge. This is guaranteed for the recurrent ResNet, but empirically we see that it is also the case for non-recurrent properly convergent ResNets. Fig. 12 demonstrates that these ResNets tend to perform a bit worse than the improperly convergent ResNets we studied in the main article.

## F RESULTS ON DIGITCLUTTER, MNIST, CIFAR-100 AND SAMPLE EFFICIENCY

Fig. 13 depicts the performance of the gradient-coupled networks on Digitclutter, CIFAR-100, MNIST, and CIFAR-10 trained with fewer samples. In none of these case, higher gradient coupling parameters systematically improve performance.

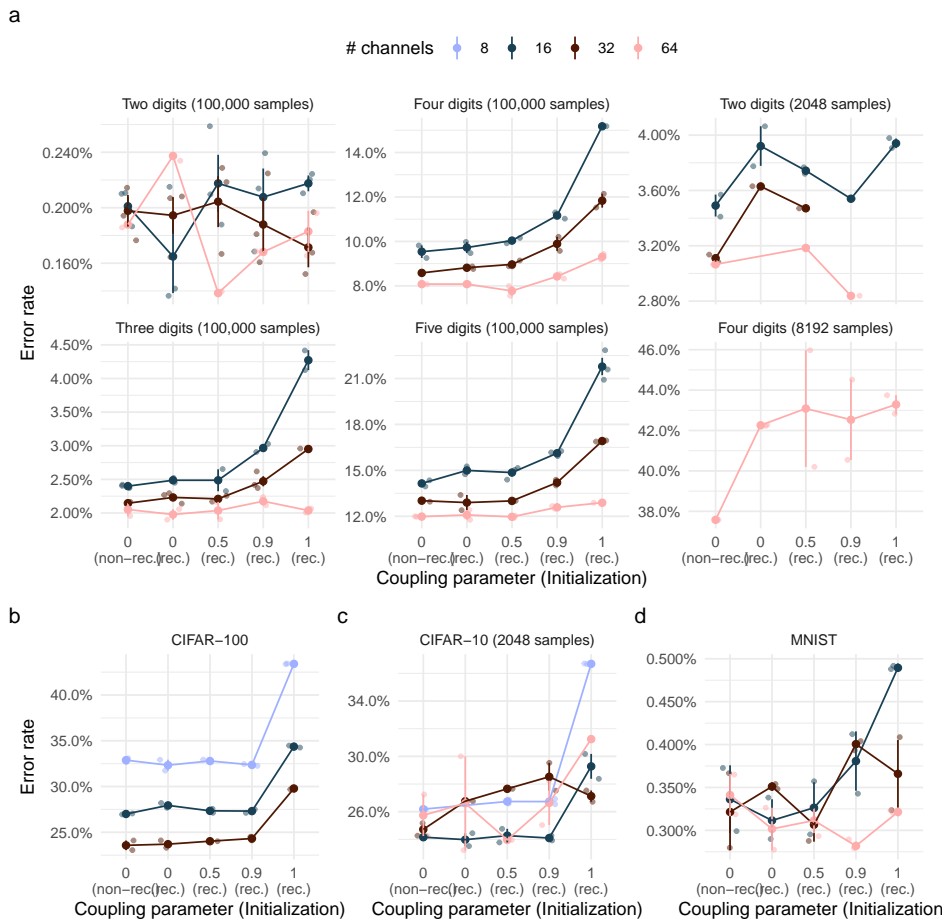

Figure 13: **a** Performance of gradient-coupled ResNets on variations of Digitclutter with a different number of overlapping digits and different size of training data. **b** Performance of gradient-coupled ResNets on CIFAR-100, CIFAR-10 with few training data, and MIST.

