# OpenReview forum: "Iterative convergent computation is not a useful inductive bias for ResNets"
_ICLR.cc/2021/Conference — Reject_

### Official Review · AnonReviewer4 · 2020-10-28
**Review for "Evidence against implicitly recurrent computations in residual neural networks"**

**Rating:** 6
**Confidence:** 4

**Review:**

This paper presents an empirical study that characterizes and quantifies the implicitly recurrent nature of residual networks (ResNets). A ResNet can be construed as a general formulation of a recurrent neural network (RNN) unfolded for a fixed number of time steps. In particular, the authors propose "soft gradient coupling," a novel way to control the degree of weight-sharing between the different residual blocks. This gives them the ability to smoothly interpolate between a "no weight sharing" scenario to a "full weight sharing scenario." Soft gradient coupling imparts the ability to share similar "computations" without necessarily sharing the same weights. They introduce metrics such as convergence, recurrence, divergence indices, and an effective parameter count to quantify "iterative" behavior numerically. Finally, they also test the impact of "iterative" computations in a ResNet trained for non-trivial visual recognition tasks.

Pros:
The problem that the authors tackle is undoubtedly interesting and useful. This is particularly true in light of a growing literature analyzing ResNets as discretized dynamical systems. The demonstration that ResNets can express iterative algorithms but do not learn such algorithms by default is intuitive and powerful. The authors use a simple toy example (a linear function) to articulate the desiderata and then work with larger datasets.

Though the manipulations to test iterative computations and implicit recurrence (early read out to determine convergence; residual block drop outs to determine recurrence; and repeated application of residual blocks to determine divergence) are not entirely novel, they are applied quite aptly. The most salient observation is that of divergence and is reminiscent of stability analysis of RNNs for vision. The notion that ResNets learn to tradeoff (and balance) feedforward vs. iterative computations is an interesting proposal.

The proposed "soft gradient coupling" scheme allows for different residual blocks to implement similar "computations" without necessarily sharing weights or changing the primary optimization problem. This is an interesting suggestion.

This paper also presents fairly extensive numerical experiments.

Cons:
The strong conclusion that ResNets do not benefit from recurrence regularization is premature, given the current set of experiments presented in this manuscript. As the authors themselves point out, "iterative computation" is an inductive bias. However, there is little reason to believe that this inductive bias is the right one for a classification problem. Have the authors tried to consider problems other than image classification? For instance, there has been recent literature on the relevance of iterative computation (visual routines) for contour detection and segmentation problems. Moreover, "accuracy" is not the only way to quantify the benefit. Have the authors tried to measure sample efficiency? i.e., can a ResNet employing iterative computations learn from fewer training samples than a non-iterative ResNet?

How does the performance benchmarking of a "fully recurrent" ResNet compare to a comparable-sized LSTM/GRU trained on this task? Or even a weight-shared ResNet? These comparisons seem to be necessary to discern if the soft gradient coupling is introducing other artificial biases.

It is unclear why the authors believed that a high degree of soft-gradient coupling would help with the divergence issue in the first place. Implementing the same computation repeatedly only converges (and stays there) given certain other properties of the transformation function applied (for instance, the spectral radius of each Residual block's Jacobian). There is quite a bit of theoretical/empirical work in the literature in this regard.

The manuscript would benefit from some discussion on theoretical results from the RNN literature that outline necessary and sufficient conditions for the forward pass of RNNs to behave like convergent dynamical systems. Given this paper's focus on iterations, convergence, and divergence, this body of work seems relevant.

Minor:
(Fig. 3a) The recurrence index was normalized, yet there are points with a recurrence index greater than 1. Is there any explanation for this?

The recurrence index measure also does not seem to add much value. (Fig 2a,b; second to left panel)

Are the values reported in Table 1., for example, point estimates? Did the authors estimate some confidence intervals on these values by running a few repeats of the experiments?

Clarity: (Pg. 2) "Encouraging iterative behavior in this way therefore does not improve the inductive bias": Is not iterative behavior *the* inductive bias?

(Fig. 3a) There must be a discussion on the non-monotonicity of these curves (especially convergence/divergence).

The paper can do with a through reformatting of the reference list to make all entries consistent in citation style (for ex: including URLs, DOIs, proper and consistent journal/conference abbreviations, etc.)

---

> ### Author Response · Authors · 2020-11-25
> **Response**
>
> Thank you for your review! We found your suggestions very helpful and lay out below how we attempted to address them in our rebuttal. Moreover, our results on the performance of the ResNets now include a measure of variation.
>
> > The strong conclusion that ResNets do not benefit from recurrence regularization is premature, given the current set of experiments presented in this manuscript. As the authors themselves point out, "iterative computation" is an inductive bias. However, there is little reason to believe that this inductive bias is the right one for a classification problem. Have the authors tried to consider problems other than image classification? For instance, there has been recent literature on the relevance of iterative computation (visual routines) for contour detection and segmentation problems. Moreover, "accuracy" is not the only way to quantify the benefit. Have the authors tried to measure sample efficiency? i.e., can a ResNet employing iterative computations learn from fewer training samples than a non-iterative ResNet?
>
> During the rebuttal, we have explored several new tasks. First of all, we found no benefit in recurrence regularization when training ResNets on CIFAR-10 with only 2048 samples. We also further explored Digitclutter, a task which involves recognizing several overlapping digits. Due to these partial occlusions, Digitclutter and related tasks have previously been demonstrated to benefit from recurrent processing (Wyatte et al., 2014; Spoerer et al., 2017). For the investigated ResNets, however, encouraging recurrent processing using higher coupling parameters did not provide a useful inductive bias.
>
> We appreciate the reviewer’s suggestion of potential experiments that may make our conclusion more convincing. We hope that the experiments we added in the rebuttal take a first step into that direction, but have also modified our language to emphasize that our conclusion, for now, are limited to the examined set of experiments. We have also added a paragraph in section 2 laying out why the Digitclutter task is particularly relevant in the context of recurrence regularization. In the future, it would be very interesting to train recurrence-regularized ResNets on a more diverse set of tasks such as image segmentation.
>
> > How does the performance benchmarking of a "fully recurrent" ResNet compare to a comparable-sized LSTM/GRU trained on this task? Or even a weight-shared ResNet? These comparisons seem to be necessary to discern if the soft gradient coupling is introducing other artificial biases.
>
> Whereas we have not trained an LSTM or GRU on this task, we note that a fully coupled ResNet corresponds to a weight-shared ResNets; if the parameters’ gradients are fully coupled, the parameters themselves will remain equal across blocks within a stage throughout the entire training. This is the reason why soft gradient coupling allows us to interpolate between an ordinary and a recurrent ResNet. It would, however, certainly be interesting to explore alternative forms of recurrence regularization to see if the soft gradient coupling is introducing any biases.
>
> > It is unclear why the authors believed that a high degree of soft-gradient coupling would help with the divergence issue in the first place. Implementing the same computation repeatedly only converges (and stays there) given certain other properties of the transformation function applied (for instance, the spectral radius of each Residual block's Jacobian). There is quite a bit of theoretical/empirical work in the literature in this regard.
>
> This is absolutely right and we have modified our language to make clear that the divergence of recurrent networks is not surprising. In addition, we have added a new set of models where the spectral radius of the residual function within each block is constrained. This allowed us to explore whether convergence provides a good inductive bias for ResNets. Our results suggest that it does not (see section 5.2 and 5.3), further supporting our conclusion that iterative convergence, under the four tasks we studied, may not be a useful inductive bias. In introducing our method of making ResNets convergent (see section 4.2 and Appendix B), we also discuss theoretical convergence results from the literature.
>
> Wyatte, D., Jilk, D. J., & O’Reilly, R. C. (2014). Early recurrent feedback facilitates visual object recognition under challenging conditions. Frontiers in Psychology, 5, 674.
>
> Spoerer, C. J., McClure, P., & Kriegeskorte, N. (2017). Recurrent Convolutional Neural Networks: A Better Model of Biological Object Recognition. Frontiers in Psychology, 8.

---

> > ### Author Response · Authors · 2020-11-25
> > **Response pt. 2**
> >
> > > The recurrence index was normalized, yet there are points with a recurrence index greater than 1. Is there any explanation for this?
> >
> > A recurrence index greater than 1 essentially indicates that dropping out earlier blocks may result in better performance than dropping out the last block. Since dropping out any block does not strongly affect the ResNet's performance, this may occasionally happen.

---

### Official Review · AnonReviewer1 · 2020-10-28
**Interesting investigation but I found the metrics and conclusion not quite convincing**

**Rating:** 5
**Confidence:** 4

**Review:**

> Summary: This paper investigates the relationship between deep ResNets and (implicitly) iterative computations. The authors introduce two main hypotheses that are at the core of the investigation: 1) whether the iterative inductive bias improves ResNet performance; and 2) whether recurrent ResNets are more parameter-efficient. The paper also proposes three metrics for studying the convergence and divergence behaviors of these networks in order to investigate this matter.

----------------

Post-rebuttal thoughts:

I would like to thank the authors for their detailed response and the revisions made to the paper. I'm updating my score to 5 as part of my concerns are satisfactorily addressed, and I wished I could have more opportunities to discuss with the authors on their response. In general, my opinion is that the authors have introduced too many "artificial" components to the study (e.g., soft gradient coupling, the convergence/divergence indices) that make me slightly dubious of how generalizable this characterization is. For example, as the authors indicated, spectral normalization creates a different phenomenon (at a cost of worse performance), but with no change to the structure itself (so unlike the soft gradient coupling), a different phenomenon could be challenging the conclusion of the paper.

My suggestion would be that the authors delve deeper into the observations here and better integrate the revisions with their original approach (e.g., the high-dimensional discussion; the spectral normalization discussion, etc.)

----------------

- I feel that in the rebuttal phase the authors made certain important new edits to the paper (e.g.,

My general opinion is that this paper investigates an interesting direction on the learning behavior of ResNets, but is still not quite ready for publication in a venue like ICLR. There is an obvious gap in related work (see my detailed comment below) on implicit deep networks; moreover, the definition of the various indices (e.g., convergence index) is also rather confusing to me. The empirical results are not strong enough evidence, in my view, to make most of the claims conclusively. I also have some doubts on the motivation for the methodology that the authors are using.

Pros:

1. Interesting direction; as the author shows, the ResNet architecture itself is expressive enough for implementing iterative computations/algorithms. So it is worthwhile to study its behavior along this trail.
2. The paper is overall written in a clear manner and the author explained their methodology well.

Cons:

1. Many arguments are too hand-wavy and I don't particularly find the metrics the authors define to analyze convergence/divergence particularly convincing. (See my comment below)
2. The experimental setup is mostly on small scales.
3. Even with the small scale setups, the experimental results don't seem conclusive enough (at least to me) to draw the conclusion that the authors were trying to claim. The verification of hypothesis 2 is especially hasty.
4. The motivation behind the soft gradient coupling is not clear to me.
5. There is a clear missing gap in the related work that I think the authors should pay attention to.

--------------------------------------

I will expand on some of the Cons above, and provide the following detailed comments/questions:

1. Again, I think it is interesting to investigate the relationship between ResNets and iterative computations. But besides the canonical, plain unrolling of the layers that the authors have looked at, *implicit models* (i.e., models that study the continuous dynamics of a layer $f$) like Neural ODEs [1] and Deep Equilibrium Models [2] (there's a ResNet version of it) are both also looking at compact recurrent networks. In particular, the deep equilibrium models especially targets the convergence (i.e., the "fixed point" of the layer), and seems to demonstrate state-of-the-art level performances. In contrast to what the authors provided in the last paragraph of Section 1, I would therefore argue (based on Neural ODEs and deep equilibrium nets) that recurrence does offer some notable advantages like constant memory cost and analytical gradients. The other related thread of work is simply the classical recurrent backprop (RBP) theories, which study the convergence of recurrent networks and how one can leverage such property for the backward pass of these networks. I found the current version of the paper did not discuss either aspect of this, which I believe is important literature that actually is on the opposite side (partially) of what the authors are trying to claim.

2. There are actually many ways that I can think of to make recurrent residual blocks converge when you infinitely repeat it. For example, with spectral normalization [3], we can simply make the Jacobian of the block have an operator norm $<1$. Then Banach fixed point theorem will guarantee convergence. Other methods are also possible (e.g., via a provably convergent optimization perspective). These are not discussed in the paper (nor are they the main focus, I guess), but this doesn't mean that ResNets do not converge in general. The authors argue that "some balance between feedforward and iterative computations might have been learned by the ResNets", but there is actually a lot of noise in the analysis... for example, the networks could be overfitting, etc. The point is, as long as you regularize the model in that direction, the ResNets could still converge.

3. One main problem that I found about this paper is its definition of the convergence/divergence indices. The "convergence" concept in this paper is constrained to look at the accuracy convergence, by which the authors look at the inverse of the AUC of the classification rate curve. But given the nature of softmax and classification task itself, I don't think a convergence in accuracy is a good "index" for measuring convergence of an architecture, which Section 3.1 looks at (for $\hat{z}_i^{(t)}$). For example, softmax is constant up to a shift of constant. And for classification of, let's say 10 objects $(x_1, \dots, x_{10})$, getting $x_1, \dots, x_5$ correct is still different from getting $x_6, \dots, x_{10}$ correct, even though they both have "50% accuracy". The paper investigates CIFAR-10, where one can achieve >94% accuracy, but in cases like ImageNet where 70% accuracy is normal, these two 50% are certainly non-convergent to me. Also, I'm assuming the entire Figure 1 is on the simple 2-dimensional linear task? Does the phenomenon in Figure 1d repeat in high dimensionality? If so, what does it look like? (My experience with this suggests that if you keep stacking the same block, the activations will eventually oscillate, if not converge, but it could differ by initialization.)

4. Some arguments are also a bit handwavy to me and I'd appreciate if the authors can expand on them. For example, in Section 3.2, the paper claims "in contrast, the skip connections encourage a ResNet to use the same representational format across blocks... [and] are therefore better aligned with the final decoder". As another example, the paper claims ResNets learn a balance between "feedforward and iterative computations". These are all intuitively reasonable arguments indeed, but considering that this is an empirical study paper, I think actually verifying these would make the paper stronger.

5. About the soft gradient coupling, doesn't this simply mix the gradients and inject more stochasticity to them? In general, would you expect (when $0 < \lambda < 1$) that just like in typical SGD, this stochasticity will be averaged out by the optimization procedure of deep networks? Since the $\tilde{\Delta}_t$ no longer fully reflect the mini-batch gradient descent direction, have you checked how the block parameters within the same stage gradually deviate from one another as you optimize the network (e.g., how does the standard deviation of $\mathbf{W}_l^{(s)}$ over all layer $l$'s in the same $s$ change over training iterations? Do they deviate or stay around? If these weights are eventually still different, why can one still consider them to be "similar" (other than the RI metric, which I find to be a debatable metric given the #3 above...)?

6. For the EPC, have you computed the EPC of an ordinary ResNet and a purely recurrent ResNet? How do their EPC look like when compared to the soft gradient coupled ResNets (e.g., $\lambda=0.5$)?

7. In Section 5.3, the paper claims that "if this is the case, we would expect soft gradient coupling to find such a solution." Why? And isn't a soft gradient coupled ResNet still a non-recurrent ResNet (in the sense that you can't simply unroll a single layer to get the output; you still need to store all parameters of the network, rather than only a single layer of it)?

--------------------------------------------

I look forward to the authors' response on my questions/comments above. I'm happy to consider adjusting my score accordingly.


[1] https://arxiv.org/abs/1806.07366
[2] https://arxiv.org/abs/1909.01377
[3] https://arxiv.org/abs/1802.05957

---

> ### Author Response · Authors · 2020-11-25
> **Response**
>
> Thank you for your review, which has helped us better integrate our work into the existing literature.
>
> > Again, I think it is interesting to investigate the relationship between ResNets and iterative computations. But besides the canonical, plain unrolling of the layers that the authors have looked at, implicit models (i.e., models that study the continuous dynamics of a layer $f$) like Neural ODEs [1] and Deep Equilibrium Models [2] (there's a ResNet version of it) are both also looking at compact recurrent networks. In particular, the deep equilibrium models especially targets the convergence (i.e., the "fixed point" of the layer), and seems to demonstrate state-of-the-art level performances. In contrast to what the authors provided in the last paragraph of Section 1, I would therefore argue (based on Neural ODEs and deep equilibrium nets) that recurrence does offer some notable advantages like constant memory cost and analytical gradients. The other related thread of work is simply the classical recurrent backprop (RBP) theories, which study the convergence of recurrent networks and how one can leverage such property for the backward pass of these networks. I found the current version of the paper did not discuss either aspect of this, which I believe is important literature that actually is on the opposite side (partially) of what the authors are trying to claim.
>
> Thank you for pointing us to these fascinating models. Though the focus of our investigation concerns whether iterative convergent behavior provides a useful inductive bias for ResNets, these methods that are more directly related to iterative methods are highly relevant in his context. We have added a paragraph in section 2 on this body of work and have clarified the particular focus of our own work.
>
> > There are actually many ways that I can think of to make recurrent residual blocks converge when you infinitely repeat it. For example, with spectral normalization [3], we can simply make the Jacobian of the block have an operator norm $<1$. Then Banach fixed point theorem will guarantee convergence. Other methods are also possible (e.g., via a provably convergent optimization perspective). These are not discussed in the paper (nor are they the main focus, I guess), but this doesn't mean that ResNets do not converge in general. The authors argue that "some balance between feedforward and iterative computations might have been learned by the ResNets", but there is actually a lot of noise in the analysis... for example, the networks could be overfitting, etc. The point is, as long as you regularize the model in that direction, the ResNets could still converge.
>
> This is a really good point and, in fact, our rebuttal includes a new set of ResNet models whose residual functions are constrained using spectral normalization. These models demonstrate, as you stated, that the ResNets converge as long as the models are regularized in that direction. However, this convergence comes at the cost of worse performance, suggesting that the divergence of ordinary ResNets may not be a mere artifact of ordinary training.

---

> > ### Author Response · Authors · 2020-11-25
> > **Response pt. 2**
> >
> > > One main problem that I found about this paper is its definition of the convergence/divergence indices. The "convergence" concept in this paper is constrained to look at the accuracy convergence, by which the authors look at the inverse of the AUC of the classification rate curve. But given the nature of softmax and classification task itself, I don't think a convergence in accuracy is a good "index" for measuring convergence of an architecture, which Section 3.1 looks at (for $\hat{z}_i^{(t)}$). For example, softmax is constant up to a shift of constant. And for classification of, let's say 10 objects $(x_1, \dots, x_{10})$, getting $x_1, \dots, x_5$ correct is still different from getting $x_6, \dots, x_{10}$ correct, even though they both have "50% accuracy". The paper investigates CIFAR-10, where one can achieve >94% accuracy, but in cases like ImageNet where 70% accuracy is normal, these two 50% are certainly non-convergent to me. Also, I'm assuming the entire Figure 1 is on the simple 2-dimensional linear task? Does the phenomenon in Figure 1d repeat in high dimensionality? If so, what does it look like? (My experience with this suggests that if you keep stacking the same block, the activations will eventually oscillate, if not converge, but it could differ by initialization.)
> >
> > We agree that our definition of the indices of iterative convergence, as based on accuracy, has certain limitations. We would argue that in some contexts, we actually only care about perturbations that affect accuracy. For instance, if the latter blocks of a ResNet only shift the final output by a constant, this does not matter if we are interested in the network’s label prediction.
> >
> > Nevertheless, we agree with the reviewer that only using accuracy is a limitation and that this should be addressed. For this purpose, we have proposed a number of alternative definitions and laid them out in section B and Fig. 6 in the Appendix. To address the example with the incompatible predictions resulting in the same accuracy, we measured accuracy with respect to the unperturbed network rather than the ground truth. Fig. 6a demonstrates that this does not change our conclusions. Thank you for this example, which we have included as motivation in section B.
> >
> > Moreover, we could also use more nuanced measures of distance. For example, instead of accuracy, we could measure crossentropy, or we could simply measure the Euclidean distance between the intermediate and the final representation. Fig. 6b and 6c visualize the effect of the perturbations defining the Convergence and Divergence Index, respectively.
> >
> > This allows us to answer the reviewer’s latter question. The fact that the Euclidean distance monotonously increases with additional evaluations of the last block suggests that for ordinary and gradient-coupled ResNets, the phenomenon in Fig. 1d indeed repeats in high dimensionality and the representation smoothly moves away from its final outcome. It is intriguing that this is different from the reviewer’s experience and it would be interesting to compare the conditions under which this trajectory begins to oscillate.
> >
> > > Some arguments are also a bit handwavy to me and I'd appreciate if the authors can expand on them. For example, in Section 3.2, the paper claims "in contrast, the skip connections encourage a ResNet to use the same representational format across blocks... [and] are therefore better aligned with the final decoder". As another example, the paper claims ResNets learn a balance between "feedforward and iterative computations". These are all intuitively reasonable arguments indeed, but considering that this is an empirical study paper, I think actually verifying these would make the paper stronger.
> >
> > We have rewritten the latter part of the introduction and hope that the revised version more clearly outlines our motivation and arguments. In particular, we would like to clarify that the intuition that ResNets learn a balance of feedforward and iterative computations arises from their behavior under the perturbations detailed in section 3.3. This intuition motivates the hypothesis that iterative convergent behavior may provide a useful inductive bias. Since for our experiments, this hypothesis seems to be wrong, we would, if anything argue that the intuition that ResNets learn this balance is misleading.

---

> > > ### Author Response · Authors · 2020-11-25
> > > **Response pt. 3**
> > >
> > > >About the soft gradient coupling, doesn't this simply mix the gradients and inject more stochasticity to them? In general, would you expect (when $0 < \lambda < 1$) that just like in typical SGD, this stochasticity will be averaged out by the optimization procedure of deep networks? Since the $\tilde{\Delta}_t$ no longer fully reflect the mini-batch gradient descent direction, have you checked how the block parameters within the same stage gradually deviate from one another as you optimize the network (e.g., how does the standard deviation of $\mathbf{W}_l^{(s)}$ over all layer $l$'s in the same $s$ change over training iterations? Do they deviate or stay around? If these weights are eventually still different, why can one still consider them to be "similar" (other than the RI metric, which I find to be a debatable metric given the #3 above...)?
> > >
> > > The reviewer raises a number of important questions here. Regarding the latter question, the average deviation between block parameters within the same stage after training decreases with higher coupling parameters (see Fig. 7c). In that sense, they are therefore more similar to each other. Here we define this deviation as the average absolute difference between the block parameters and their mean across blocks within the same stage.
> > >
> > > Regarding the former question, we would expect the bias introduced by soft gradient coupling not to be averaged out over multiple samples. After all, soft gradient coupling -- consistent, throughout all samples of SGD -- is biased towards aligning the gradients of the parameters across blocks within a stage and therefore encourages the optimization procedure to find more recurrent minima.
> > >
> > > > For the EPC, have you computed the EPC of an ordinary ResNet and a purely recurrent ResNet? How do their EPC look like when compared to the soft gradient coupled ResNets (e.g., $\lambda=0.5$)?
> > >
> > > Figure 7 (in the appendix) depicts the EPC plotted against the raw parameter count. The EPC of fully recurrent and non-recurrent ResNets is roughly equal to their raw parameter count and decreases for higher intermediate coupling parameters. Note that our revised manuscript has shifted the focus away from the effective parameter count and we have therefore moved this section to the appendix.
> > >
> > > > In Section 5.3, the paper claims that "if this is the case, we would expect soft gradient coupling to find such a solution." Why?
> > >
> > > Our language here was too strong and we have modified it accordingly. We would expect that soft gradient coupling might be able to find such a solution because the coupled gradients shift the training dynamics towards more recurrent solutions without the parameter space being constrained to a, perhaps overly restrictive, space of fully recurrent networks.
> > >
> > > > And isn't a soft gradient coupled ResNet still a non-recurrent ResNet (in the sense that you can't simply unroll a single layer to get the output; you still need to store all parameters of the network, rather than only a single layer of it)?
> > >
> > > If $\lambda<1$, this is true. However, as the EPC demonstrates, in a softly coupled network, the parameters of the individual blocks within a stage are less spread out, which may allow us to more easily find a partly recurrent structure expressing this computation. However, this is merely speculative at the moment and we have therefore shifted the focus of the article away from this point. Instead, we have decided to focus on the more concrete question whether iterative convergent behavior provides a useful inductive bias.

---

### Official Review · AnonReviewer3 · 2020-10-28

**Rating:** 5
**Confidence:** 3

**Review:**


Summary:

This paper investigates the extent to which the computations implemented by an optimized ResNet resemble those of a recurrent network. They develop new tools to study this question, and find evidence that ResNet performance is *hurt* when it is forced to act like a specific kind of RNN.

Strengths:

I am excited that the authors are re-examining the assumptions of now classic work likening RNNs to ResNets. They develop elegant tools for continuously transforming between the two, and compare performance on several image classification tasks.

Weaknesses:

I'm not totally convinced by the author's pitch for inverse problems. This idea of the visual system acting as a generative model is still far from worked out, and I'd prefer the authors hedge their language on it. There's also other tasks that are more closely linked to recurrent processing than the ones studied here. For example, I recommend the authors experiment with Pathfinder or cABC of [1], which are solved in fewer samples by recurrent networks vs. feedforward models. This would allow you to plot the amount of gradient coupling on one axis, and the number of samples need to solve (e.g.) Pathfinder on the other axis, which I think would be very elegant.

I appreciate the authors laying out their hypotheses like they did, but I found the language to be indirect. Is there a simpler way to motivate these hypotheses?

Figure 1 is difficult to understand. Are you plotting activities against each other? What do the different dots represent for the feedforward model? Is the interaction in (d) meaningful or is this just by happenstance, and the divergence from x is the meaningful quality.

Why ResNet-104?

When dropping ResNet blocks, how do you deal with the subsampling between layers? When you drop out the first layer do you also drop out the max pooling at that layer? Is there any chance that these distinctions in computations and resolutions between blocks that you're dropping could bias your observed results?

Regarding the Divergence index, the authors should review [2]. I don't think a high divergence index necessarily means that the ResNet isn't learning the function of an RNN — only that the learned function is not stable, which makes sense given ResNet hyperparams. This paper suggests that if you change the model nonlinearities to globally contractive ones like tanh or sigmoid (or use their algorithm) you'll control this problem.

The gradient coupling is forcing a fixed combination between the gradients of successive layers. But gated RNNs are standard for recurrent vision models, and these do not have such a constraint. Is it possible that the ResNet without shared weights is learning a the function of a gated RNN rather than the vanilla RNNs that you're comparing to here?

"Our findings also suggest, however, that deep feedforward computations may not be characterized as iterative refinement on a latent representation, but, at most, as non-iterative refinement on this representation." How do you refine non-iteratively/incrementally? More generally, I felt like the authors overloaded "iterative" and the manuscript would benefit from a more careful treatment of the exact computations they're /referring to. Give concrete examples.

Are you comparing ResNets to RNNs or iterative algorithms (as are alluded to in the intro)? I am confused by the motivation here, which changes from paragraph to paragraph.

[1] Kim et al. Disentangling neural mechanisms for perceptual grouping. ICLR 2020.
[2] Linsley et al. Stable and expressive recurrent vision models. NeurIPS 2020.

---

> ### Author Response · Authors · 2020-11-25
> **Response pt. 1**
>
> Thank you for your helpful review. Your questions helped us a lot in revising the explanations in our manuscript. Below we directly answer these questions and detail how we attempted to address the issues you have raised.
>
> > This idea of the visual system acting as a generative model is still far from worked out, and I'd prefer the authors hedge their language on it.
>
> We agree with this and have modified our language accordingly.
>
> > There's also other tasks that are more closely linked to recurrent processing than the ones studied here. For example, I recommend the authors experiment with Pathfinder or cABC of [1], which are solved in fewer samples by recurrent networks vs. feedforward models. This would allow you to plot the amount of gradient coupling on one axis, and the number of samples need to solve (e.g.) Pathfinder on the other axis, which I think would be very elegant.
>
> Thank you for suggesting these datasets. Evaluating the different models on these tasks would indeed be interesting and provide important insights into whether recurrence regularization can provide a useful inductive bias for ResNets. Since these images, at 300 x 300 pixels, are much larger than the comparably small datasets we have considered so far, two weeks were not enough time to adapt the ResNets to this larger task.
>
> However, we have included another task, which is also more closely linked to recurrent processing than Cifar-10 and MNIST: Digitclutter consists of a number of partially occluded digits. Recognizing occluded stimuli often requires recurrent processing in humans (Wyatte et al., 2014) and convolutional neural networks have been shown to benefit from recurrent connections under this task (Spoerer et al., 2017). We have significantly extended our experiments using this dataset, examining versions ranging from two to five partially occluded digits. For all these tasks, recurrence regularization did not improve performance. We now present these results in the main text (section 5.3) and have added the paragraph ‘Inductive bias of recurrent operations on visual tasks’ to section 2 in order to motivate their significance. This paragraph also includes the relevant literature on Pathfinder and cABC.
>
> We agree that examining the relationship between recurrence regularization and performance on Pathfinder and cABC would be very interesting. We hope that our experiments on Digitclutter provide a better intuition for the ResNets’ performance on tasks which have been linked to recurrent processing.
>
> > I appreciate the authors laying out their hypotheses like they did, but I found the language to be indirect. Is there a simpler way to motivate these hypotheses?
> "Our findings also suggest, however, that deep feedforward computations may not be characterized as iterative refinement on a latent representation, but, at most, as non-iterative refinement on this representation." How do you refine non-iteratively/incrementally? More generally, I felt like the authors overloaded "iterative" and the manuscript would benefit from a more careful treatment of the exact computations they're /referring to. Give concrete examples.
> Are you comparing ResNets to RNNs or iterative algorithms (as are alluded to in the intro)? I am confused by the motivation here, which changes from paragraph to paragraph.
>
> Thank you for raising this issue. We have rewritten the latter part of the introduction and hope that our revised motivation is stated more clearly. More specifically, our investigation was motivated by the observation that the feedforward computations within ResNets have certain similarities to the recursive operations of an iterative method: throughout the layers the representation is gradually refined, slowly approaching its final state. It has previously been proposed that this iterative refinement may be part of the reason for ResNets’ good performance on many computer vision tasks (Jastrzębski et al., 2018). This suggests that iterative convergent behavior may be a useful inductive bias for ResNets. In our article, we aim to investigate whether this is the case.
>
> We appreciate the reviewer detailing why the terminology and motivation of the original manuscript had been confusing. We have revised the corresponding parts of the manuscript and we hope that our motivation and terminology are now more clearly laid out.

---

> > ### Author Response · Authors · 2020-11-25
> > **Response pt. 2**
> >
> > > Figure 1 is difficult to understand. Are you plotting activities against each other? What do the different dots represent for the feedforward model? Is the interaction in (d) meaningful or is this just by happenstance, and the divergence from x is the meaningful quality.
> >
> > Figure 1 depicts the two-dimensional estimates of the output according to the different considered algorithms. In the case of the feedforward model, each dot represents the readout after a certain intermediate stage. Since the subsequent layers are not aligned to each other, this early readout does not result in a useful estimate, as is apparent from the figure. This is in contrast to the intermediate estimate from the residual networks, which smoothly approach their final output. By interaction, do you mean that the two trajectories in (d) are roughly orthogonal to each other? In that case, this is pure happenstance; rather, the fact that both trajectories diverge from x is relevant. Thank you for raising these issues; we have attempted to explain the figure more clearly in the main text as well as the figure legend.
> >
> > > Why ResNet-104?
> >
> > We chose 16 residual blocks per layer, because we considered this a representative depth for a typical ResNet. For example, He et al. (2016) consider a similarly deep ResNet-110. We expect these results to be robust across different depths. We would be happy to train network instances with a different number of residual blocks for the camera-ready version of the paper. (Note that we should have referred to the considered network as ResNet-101 instead of ResNet-104 and have now corrected this in the manuscript.)
> >
> > > When dropping ResNet blocks, how do you deal with the subsampling between layers? When you drop out the first layer do you also drop out the max pooling at that layer? Is there any chance that these distinctions in computations and resolutions between blocks that you're dropping could bias your observed results?
> >
> > The subsampling between layers is implemented as part of an additional residual block, which is never removed as part of the perturbations. We have added a sentence in section A.3 to make this clearer. Note that the subsampling does not work by max pooling, but by direct subsampling with stride 2, just as in the original ResNet implementation.
> >
> > > Regarding the Divergence index, the authors should review [2]. I don't think a high divergence index necessarily means that the ResNet isn't learning the function of an RNN — only that the learned function is not stable, which makes sense given ResNet hyperparams. This paper suggests that if you change the model nonlinearities to globally contractive ones like tanh or sigmoid (or use their algorithm) you'll control this problem.
> >
> > We agree that a high divergence index does not necessarily mean that the ResNets is not learning a recurrent function. We appreciate the reviewer raising this issue and hope this is more clear in the reviewed manuscript. Instead, the Divergence Index is intended to be a measure of whether the learned function is stable, i. e. convergent. Indeed, the reviewer raises an important point here: even though higher coupling parameters generally lead to a lower Divergence Index, the considered recurrent and non-recurrent ResNets are all not convergent. Since we aim to study not only the impact of iterative, but also convergent behavior on the network’s inductive bias, this was a limitation. To address this, we have now introduced spectral normalization to define convergent ResNets in the revised manuscript. Our method is based on the upper bound on the Lipschitz constant of a convolutional neural network as introduced by Yoshida et al. (2017), but a similar network could be defined using the method by Linsley et al. (2020). We discuss their relationship in the last paragraph of section 2.
> >
> > > The gradient coupling is forcing a fixed combination between the gradients of successive layers. But gated RNNs are standard for recurrent vision models, and these do not have such a constraint. Is it possible that the ResNet without shared weights is learning the function of a gated RNN rather than the vanilla RNNs that you're comparing to here?
> >
> > That is an interesting question. We think it is, in principle, possible for a ResNets to learn the function of a gated RNN or at least approximate it very well on the data distribution. This could be even more easily implemented by a highway network, which, depending on its implementation, is actually its non-recurrent generalization. Coupling the gradients of a highway network would allow us to interpolate between non-recurrent highway networks and gated RNNs. This could induce a useful inductive bias and would certainly constitute an interesting investigation for the future.

---

> > > ### Author Response · Authors · 2020-11-25
> > > **Response pt. 3: References**
> > >
> > > Jastrzębski, S., Arpit, D., Ballas, N., Verma, V., Che, T., & Bengio, Y. (2017). Residual Connections Encourage Iterative Inference. https://arxiv.org/abs/1710.04773v2
> > >
> > > Wyatte, D., Jilk, D. J., & O’Reilly, R. C. (2014). Early recurrent feedback facilitates visual object recognition under challenging conditions. Frontiers in Psychology, 5, 674.
> > >
> > > Spoerer, C. J., McClure, P., & Kriegeskorte, N. (2017). Recurrent Convolutional Neural Networks: A Better Model of Biological Object Recognition. Frontiers in Psychology, 8.
> > >
> > > Yoshida, Y., & Miyato, T. (2017). Spectral norm regularization for improving the generalizability of deep learning. ArXiv Preprint ArXiv:1705.10941.
> > >
> > > Linsley, D., Ashok, A. K., Govindarajan, L. N., Liu, R., & Serre, T. (2020). Stable and expressive recurrent vision models. ArXiv:2005.11362 [Cs]. http://arxiv.org/abs/2005.11362

---

### Official Review · AnonReviewer2 · 2020-10-29
**Concerns about motivations and results**

**Rating:** 5
**Confidence:** 4

**Review:**

## Paper Summary

This paper studies the correspondence between residual networks and iterative algorithms that repeat computations and converge to a solution. The authors suggest that residual networks can in principle implement such iterative algorithms and experimentally show that networks trained in practice do not naturally learn them. They also define three indices to quantify the degree to which a ResNet shows properties of iterative algorithms. Finally, they show that while soft gradient coupling across layers within stages can ensure that learned ResNets behave more like iterative algorithms, this does not appear to provide a useful inductive bias for image classification tasks.

## Strengths

Studying the nature of programs that are learned by neural networks of various architectures is an interesting and important research problem. This paper makes a contribution to it by examining the extent to which ResNets implement algorithms similar to iterative solvers.

The authors define numerical indices to formalize the criteria for "iterative-ness" that they are looking for, which are useful for comparisons.

The paper contains a negative result about the utility of forcing iterative behavior on ResNets using the proposed gradient coupling trick. This negative result may be useful to researchers interested in similar ideas in the future.

## Weaknesses

The motivation for this paper is somewhat weak, or at least weakly justified. The authors define an iterative method/algorithm as one that uses repeated iterations and convergences to a solution. It is then hypothesized that such behavior might be a good inductive bias for neural networks, but it is not discussed why this might be expected. After all, iterative algorithms are designed to converge after a (non-fixed) number of steps, and neural nets are not. I think that either the motivation should be justified better, or it is simply a question without a strong motivation (this doesn't necessarily make it unimportant, just less important).

Moreover, if we'd like the outputs of the neural network to be "stable" for reasons other than metrics like accuracy, there are certain methods and lines of research on this subject, such as Ciccone et al. cited by the authors. Those works already claim that computations learned by ResNets are not stable, and suggest methods to make them so. Doesn't that make the question investigated (whether ResNets learn iterative convergent behavior) in this paper somewhat redundant?

Finally, the negative experimental results are interesting but I think they need to be stronger to be convince a reader that this is a result that can be expected to generalize. Due to the simplicity of the datasets (and no confidence intervals on the numbers in Table 1), evidence for the negative impact of encouraging iterative convergent behavior on performance is still preliminary. More datasets and tasks of higher complexity will certainly help here.

## Review Summary

Since the paper lacks strong motivations, clear significance and highly convincing results, I am currently unable to recommend an acceptance. If the authors can elaborate on their motivations and discuss them in light of related work as mentioned above, I am willing to reconsider my score.

## After Author Response

I think the changes to the paper have improved it. In particular, the reference to Spoerer et al. gives more weight to the motivations of the paper. I'm increasing my score slightly as a result.
However, the relation to prior work remains hazy. The model of Ciccone et al., for example, has shared weights, stability over increased depth, and still performs as well as ResNets generally. Doesn't this go against the conclusions of this paper?

---

> ### Author Response · Authors · 2020-11-25
> **Response**
>
> Thank you for your insightful review. In our rebuttal, we have attempted to address the points you have raised by running new experiments and revising our manuscript.
>
> > The motivation for this paper is somewhat weak, or at least weakly justified. The authors define an iterative method/algorithm as one that uses repeated iterations and convergences to a solution. It is then hypothesized that such behavior might be a good inductive bias for neural networks, but it is not discussed why this might be expected. After all, iterative algorithms are designed to converge after a (non-fixed) number of steps, and neural nets are not. I think that either the motivation should be justified better, or it is simply a question without a strong motivation (this doesn't necessarily make it unimportant, just less important).
>
> Thank you for pointing this out. Our investigation was motivated by the observation that the feedforward computations within ResNets have certain similarities to the recursive operations of an iterative method: throughout the layers the representation is gradually refined, slowly approaching its final state. It has previously been proposed that this iterative refinement may be part of the reason for ResNets’ good performance on many computer vision tasks (Jastrzębski et al., 2018). This suggests that iterative convergent behavior may be a useful inductive bias for ResNets. We have rewritten the latter part of the introduction and hope that our revisions have clarified our motivation.
>
> > Moreover, if we'd like the outputs of the neural network to be "stable" for reasons other than metrics like accuracy, there are certain methods and lines of research on this subject, such as Ciccone et al. cited by the authors. Those works already claim that computations learned by ResNets are not stable, and suggest methods to make them so. Doesn't that make the question investigated (whether ResNets learn iterative convergent behavior) in this paper somewhat redundant?
>
> Thank you for raising this issue, it has helped us refine the presentation of our motivation as we have detailed above. In particular, we have emphasized our investigation into whether iterative convergent behavior is a useful inductive bias for ResNets. Though higher gradient coupling parameters also increase the Convergence Index and decrease the Divergence Index, this method largely focuses on making ResNets more iterative. We therefore complement our recurrence regularization by a convergence regularization using prior work on Lipschitz bounds on convolutional neural networks (Yoshida et al., 2017). Section 4.2 defines our new convergent ResNets and sections 5.2 and 5.3 summarise our findings on their performance. To summarise the findings, this convergence regularization negatively impacts performance, as well, providing further evidence that ResNets may not benefit from iterative convergent behavior.
>
> > Finally, the negative experimental results are interesting but I think they need to be stronger to be convince a reader that this is a result that can be expected to generalize. Due to the simplicity of the datasets (and no confidence intervals on the numbers in Table 1), evidence for the negative impact of encouraging iterative convergent behavior on performance is still preliminary. More datasets and tasks of higher complexity will certainly help here.
>
> We agree that it is difficult to demonstrate this result and have stated more carefully that our negative results are limited to the tasks we have considered. Thank you for proposing additions that may convince a reader of the generality of the results. To convey the performance variation across several training instances, we now plot performance in a scatterplot, using mean and standard deviation as summary statistics (Figure 4). This demonstrates that performance is quite consistent across several instances. We have also added more datasets to our investigation. First of all, our results replicate on CIFAR-100 (see Fig. 13).
>
> Moreover, the Digitclutter dataset requires the ResNets to recognize partially occluded digits. Such tasks involve recurrent processing in humans (Wyatte et al., 2014) and benefit from recurrent connections in certain convolutional neural networks (Spoerer et al., 2017). If there are datasets for which recurrence regularization provides an advantage for ResNets, we may therefore expect Digitclutter to be among them. We have significantly extended our results on Digitclutter, demonstrating that recurrence regularization does not provide a better inductive bias for datasets with a wide range of complexity. We now present parts of these results in the main text (section 5.3) and have added the paragraph ‘Inductive bias of recurrent operations on visual tasks’ to section 2 in order to motivate their significance. We hope that these experiments will make our findings more generally applicable.

---

> > ### Author Response · Authors · 2020-11-25
> > **Response pt. 2: References**
> >
> > Jastrzębski, S., Arpit, D., Ballas, N., Verma, V., Che, T., & Bengio, Y. (2017). Residual Connections Encourage Iterative Inference. https://arxiv.org/abs/1710.04773v2
> >
> > Wyatte, D., Jilk, D. J., & O’Reilly, R. C. (2014). Early recurrent feedback facilitates visual object recognition under challenging conditions. Frontiers in Psychology, 5, 674.
> >
> > Spoerer, C. J., McClure, P., & Kriegeskorte, N. (2017). Recurrent Convolutional Neural Networks: A Better Model of Biological Object Recognition. Frontiers in Psychology, 8.
> >
> > Yoshida, Y., & Miyato, T. (2017). Spectral norm regularization for improving the generalizability of deep learning. ArXiv Preprint ArXiv:1705.10941.

---

### Author Response · Authors · 2020-11-25
**General remarks**

We would like to thank the reviewers for their insightful questions and helpful suggestions. The reviews have helped us significantly improve our manuscript during the rebuttal. We are glad that the reviewers found the topic of the article to be important and interesting and generally considered our proposed indices to be useful.

We respond to each reviewer individually, but would like to give an overview over the most important changes to the manuscript:

All reviewers noted that they found the motivation to be presented overly complicated, which made the results more difficult to understand. We have now attempted to lay out the motivation significantly simpler. In particular, the previous observations that a ResNet shares certain properties with an iterative method have motivated the hypothesis that ResNets approximate iterative methods. Iterative convergent behavior may therefore provide a useful inductive bias for ResNets. Iterative methods are characterized by two properties, iteration and convergence, that we manipulate in ResNets via soft gradient coupling and spectral normalization.

All reviewers suggested that it was not clear why we expected soft gradient coupling to make ResNets more convergent. We clarified that it is indeed unsurprising that the recurrent ResNets are divergent and have now included a new method of convergence regularization. This method results in a weaker performance of our trained ResNets, suggesting that convergent behavior may not be a useful inductive bias.

Finally, all reviewers suggested we explore more tasks. For our rebuttal, we added experiments on CIFAR-100. Moreover, we more extensively evaluated the performance of gradient-coupled ResNets on Digitclutter. In this task, the network must recognize the identity of a number of digits, which partially occlude each other. This task and related versions have been demonstrated to benefit from recurrent processing in humans (Wyatte et al., 2014) and artificial algorithms (Spoerer et al., 2017).

Once again, we would like to thank the reviewers for their work and look forward to a potential further discussion of our findings.



Wyatte, D., Jilk, D. J., & O’Reilly, R. C. (2014). Early recurrent feedback facilitates visual object recognition under challenging conditions. Frontiers in Psychology, 5, 674.

Spoerer, C. J., McClure, P., & Kriegeskorte, N. (2017). Recurrent Convolutional Neural Networks: A Better Model of Biological Object Recognition. Frontiers in Psychology, 8.

---

### Decision · Program_Chairs · 2021-01-07
**Final Decision**

**Decision:**

Reject

**Comment:**

This work provides evidence against the hypothesis that ResNets implement iterative inference, or that iterative convergent computation is a good inductive bias to have in these models. The reviewers indicate that they think this hypothesis is interesting and relevant to the ICLR community, but they do not find the current work sufficiently convincing. Both theoretically and experimentally the paper does not fully demonstrate the claim that iterative inference is not useful in ResNets, and the reviewers are unanimous in their recommendation to reject the paper until the evidence for this claim is strengthened.